# Identifiable Shared Component Analysis of Unpaired Multimodal Mixtures

**Subash Timilsina**
School of EECS
Oregon State University
Corvallis, OR 97331
timilsis@oregonstate.edu

**Sagar Shrestha**
School of EECS
Oregon State University
Corvallis, OR 97331
shressag@oregonstate.edu

**Xiao Fu**
School of EECS
Oregon State University
Corvallis, OR 97331
xiao.fu@oregonstate.edu

## Abstract

A core task in multi-modal learning is to integrate information from multiple feature spaces (e.g., text and audio), offering modality-invariant essential representations of data. Recent research showed that, classical tools such as *canonical correlation analysis* (CCA) provably identify the shared components up to minor ambiguities, when samples in each modality are generated from a linear mixture of shared and private components. Such identifiability results were obtained under the condition that the cross-modality samples are aligned/paired according to their shared information. This work takes a step further, investigating shared component identifiability from multi-modal linear mixtures where cross-modality samples are unaligned. A distribution divergence minimization-based loss is proposed, under which a suite of sufficient conditions ensuring identifiability of the shared components are derived. Our conditions are based on cross-modality distribution discrepancy characterization and density-preserving transform removal, which are much milder than existing studies relying on independent component analysis. More relaxed conditions are also provided via adding reasonable structural constraints, motivated by available side information in various applications. The identifiability claims are thoroughly validated using synthetic and real-world data.

## 1 Introduction

The same data entities can often be represented in different feature spaces (e.g., audio, text and image), due to the variety of sensing modalities or domains. Learning common latent components of data from multiple modalities is well-motivated in representation learning. The shared components are considered modality-invariant essential representations of data, which can often enhance performance of downstream tasks by shedding modality-specific noise [1–4] and avoiding over-fitting [5–7].

A prominent theoretical aspect of shared component learning lies in *identifiability* of the components of interest. The literature posed an intriguing theoretical question [1, 2, 8]: If every modality of data is represented by a linear mixture of shared and private components with an unknown mixing system, are the shared components identifiable (up to acceptable ambiguities)? Such component identification problems are often nontrivial due to the ill-posed nature of any linear mixture model (see, e.g., [9–14]). Interestingly, the work [1] showed that using the classical *canonical correlation*

---

38th Conference on Neural Information Processing Systems (NeurIPS 2024).

*analysis* (CCA) provably find the shared components up to rotation and scaling. In fact, shared component identification from multimodal/multiview linear mixtures were considered in various contexts (see, e.g., [15–18]), although some of these works did not model private components. The identifiability results in [1, 2] were generalized to nonlinear mixture models as well [4, 19]. The shared component identification perspective was also related to the success of representation learning in self-supervised learning (SSL) [5–7].

Nonetheless, the treatment in [1, 2] and the related works [15–17] all assumed that the cross-modality data are aligned (i.e., paired) according to their shared components. In many applications, such as cross-language information retrieval [20–22], domain adaptation [23–25], and biological data translation [26, 27], aligned cross-modality data are hard to acquire, if not outright unavailable. A natural question is: *When the multimodal linear mixtures are unaligned, can the shared latent components still be provably identified under reasonably mild conditions?*

**Existing Studies.** Theoretical characteristics of unaligned multimodal learning were studied under various settings. The work [28] considered a case where one modality is a linear transform of another modality, and showed that the linear transformation is potentially identifiable. The recent work [29] extended this model to a nonlinear transform setting. However, these works did not consider *latent* component models—yet the latter are more versatile in many ways, e.g., facilitating one-to-many cross-domain translations [30, 31]. The work [32] considered unaligned mixtures of shared and private components, but the assumptions (e.g., the availability of a large amount of modalities) to ensure identifiability may not be easy to satisfy. The most related work is perhaps [8]. But their approach also relied on somewhat stringent assumptions, e.g., that all the latent components are element-wise statistically independent with at most one component being Gaussian. This is because their procedure had to invoke the classical *independent component analysis* (ICA) [33].

**Contributions.** In this work, we provide a suite of sufficient conditions under which the shared components can be provably identified from unaligned multimodal linear mixtures up to reasonable ambiguities. The model and identification problem are referred to as *unaligned shared component analysis* (unaligned SCA) in the sequel.

*(i) An Identifiable Learning Loss for Unaligned SCA.* We propose to tackle the unaligned SCA problem by matching the probability distributions of linearly embedded multi-modal data. We show that under reasonable conditions, the linear transformations identifies the shared components up to the same ambiguities as those in the aligned case [1, 2]. The conditions are considerably milder compared to the existing unaligned SCA work [8].

*(ii) Enhanced Identifiability via Structural Constraints.* We come up with two types of structural constraints, motivated by available side information in applications, to further relax the identifiability conditions. Specifically, we look into cases where the multi-modal data have similar linear mixing systems and cases where a few cross-domain aligned samples available. We show that by adding constraints accordingly, unaligned SCA are identifiable under much milder conditions.

Our contributions primarily lie in identifiability analysis. Nonetheless, we also show the usefulness of our results in real-world applications, namely, *cross-lingual word retrieval*, *genetic information alignment* and *image data domain adaptation*. Particularly, it shows that our succinct multimodal linear mixture model can effectively post-process outputs of pre-trained encoders, e.g., those in [34, 35], to improve data representations and enhance downstream task performance.

**Notation.** Notation definitions can be found in Appendix A.

## 2 Background

**Generative Model of Interest.** Following the classical settings in [1, 2, 15, 16, 18], we consider modeling the multi-modal data as linear mixtures. More specifically, we adopt the model in [1, 2]that splits the latent representation of data into shared components and private components:

$$\boldsymbol{x}^{(q)} = \boldsymbol{A}^{(q)}\boldsymbol{z}^{(q)}, \quad \boldsymbol{z}^{(q)} = [\boldsymbol{c}^\top, (\boldsymbol{p}^{(q)})^\top]^\top, \ q = 1, 2, \tag{1}$$

where $\boldsymbol{x}^{(q)} \in \mathbb{R}^{d^{(q)}}$ represents the data from the $q$th modality, $\boldsymbol{z}^{(q)} \in \mathbb{R}^{d_{\mathrm{C}}+d_{\mathrm{P}}^{(q)}}$ represents the corresponding latent code, $\boldsymbol{c} \in \mathbb{R}^{d_{\mathrm{C}}}$ and $\boldsymbol{p}^{(q)} \in \mathbb{R}^{d_{\mathrm{P}}^{(q)}}$ stand for the shared components and the private components, respectively. The data $\boldsymbol{x}^{(q)}$'s are assumed to be zero-mean, which can be enforced by

centering. Note that the positions of $\boldsymbol{c}$ and $\boldsymbol{p}_q$ are not necessarily arranged as $[\boldsymbol{c}^\top, (\boldsymbol{p}^{(q)})^\top]^\top$ (more generally, $\boldsymbol{z}^{(q)} = \boldsymbol{\Pi}^{(q)}[\boldsymbol{c}^\top, (\boldsymbol{p}^{(q)})^\top]^\top$ with an unknown permutation matrix $\boldsymbol{\Pi}^{(q)}$). However, the representation in (1) is without loss of generality as one can define $\boldsymbol{A}^{(q)} := \boldsymbol{A}^{(q)}(\boldsymbol{\Pi}^{(q)})^\top$ to reach the representation in (1). For all the domains, we have

$$\boldsymbol{c} \sim \mathbb{P}_{\boldsymbol{c}}, \quad \boldsymbol{p}^{(q)} \sim \mathbb{P}_{\boldsymbol{p}^{(q)}}, \tag{2}$$

where $\mathbb{P}_{\boldsymbol{c}}$ and $\mathbb{P}_{\boldsymbol{p}^{(q)}}$ represent the distributions of the shared components and the domain-private components, respectively. Under (1), the two different range spaces $\mathrm{range}(\boldsymbol{A}^{(q)})$ for $q = 1, 2$ represent two feature spaces. Then latent $\boldsymbol{p}^{(q)}$ further distinguishes the modalities and often has interesting physical interpretation. For example, some vision literature use $\boldsymbol{c}$ to model "content" and $\boldsymbol{p}^{(q)}$ "style" of the images [31, 36]. In cross-lingual word embedding retrieval [2], $\boldsymbol{c}$ represents the semantic meaning of the words, while $\boldsymbol{p}^{(q)}$ represents the language-specific components. The goal of SCA boils down to finding linear operators to recover $\boldsymbol{c}$ to a reasonable extent.

**Aligned SCA: Identifiability of CCA and Extensions.** Learning $\boldsymbol{c}$ without knowing $\boldsymbol{A}^{(q)}$ is a typical component analysis problem. Learning latent components from *linear mixture models* (LMMs) like $\boldsymbol{x} = \boldsymbol{A}\boldsymbol{z}$ lacks identifiability in general, due to the bilinear nature of the models. This is because one can find an infinite number of invertible matrices $\boldsymbol{B}$ such that $\boldsymbol{x} = \boldsymbol{A}\boldsymbol{B}\boldsymbol{B}^{-1}\boldsymbol{z}$. Then, both $(\boldsymbol{A}, \boldsymbol{z})$ and $(\boldsymbol{A}\boldsymbol{B}, \boldsymbol{B}^{-1}\boldsymbol{z})$ can fit to the data $\boldsymbol{x}$, making the problem ill-posed in terms of solution uniqueness; see, e.g., [9, 37] and more discussions in Sec. 5. Nonetheless, the works [1, 2] studied the identifiability of $\boldsymbol{c}$ under the model (1), using the assumption that the cross-modality samples share the same $\boldsymbol{c}$ are aligned. In particular, [1] formulated the $\boldsymbol{c}$-identification problem as a CCA problem:

$$\underset{\{\boldsymbol{Q}^{(q)}\}_{q=1}^2}{\mathrm{minimize}} \quad \mathbb{E}\left[\left\|\boldsymbol{Q}^{(1)}\boldsymbol{x}^{(1)} - \boldsymbol{Q}^{(2)}\boldsymbol{x}^{(2)}\right\|_2^2\right] \tag{3a}$$

$$\text{subject to} \quad \boldsymbol{Q}^{(q)}\mathbb{E}\left[\boldsymbol{x}^{(q)}(\boldsymbol{x}^{(q)})^\top\right](\boldsymbol{Q}^{(q)})^\top = \boldsymbol{I} \quad q = 1, 2, \tag{3b}$$

where $\boldsymbol{Q}^{(q)} \in \mathbb{R}^{d_{\mathrm{C}} \times d^{(q)}}$. The expectation in (3a) is taken from the joint distribution of the *aligned pairs* $\mathbb{P}_{\boldsymbol{x}^{(1)}, \boldsymbol{x}^{(2)}}$, where every pair $(\boldsymbol{x}^{(1)}, \boldsymbol{x}^{(2)})$ shares the same $\boldsymbol{c}$. The formulation aims to find $\boldsymbol{Q}^{(q)}$ such that the transformed representations of the aligned pairs $\boldsymbol{Q}^{(1)}\boldsymbol{x}^{(1)}$ and $\boldsymbol{Q}^{(2)}\boldsymbol{x}^{(2)}$ are equal. In [1], it was shown that

$$\widehat{\boldsymbol{Q}}^{(q)}\boldsymbol{x}^{(q)} = \boldsymbol{\Theta}\boldsymbol{c} \tag{4}$$

under mild conditions (see Appendix E.1 for details), where $(\widehat{\boldsymbol{Q}}^{(1)}, \widehat{\boldsymbol{Q}}^{(2)})$ is an optimal solution of the CCA formulation and $\boldsymbol{\Theta}$ is a certain non-singular matrix. Eq. (4) means that $\widehat{\boldsymbol{Q}}^{(q)}$ finds the range space where $\boldsymbol{c}$ lives in, i.e., $\mathrm{range}(\boldsymbol{A}_{1:d_{\mathrm{C}}}^{(q)})$ under our notation.

**Unaligned SCA: Existing Result and Theoretical Gap.** The work in [8] studied the identifiability of $\boldsymbol{c}$ under (1) when $\boldsymbol{x}^{(1)}$ and $\boldsymbol{x}^{(2)}$ are *unaligned*. Their approach works under the condition *that the elements of $\boldsymbol{z}^{(q)} = [\boldsymbol{c}^\top, (\boldsymbol{p}^{(q)})^\top]^\top$ are mutually statistically independent.* There, $\widehat{\boldsymbol{z}}^{(q)} = \boldsymbol{\Pi}^{(q)}\boldsymbol{\Sigma}^{(q)}\boldsymbol{z}^{(q)}$ is assumed to have been estimated by ICA, where $\boldsymbol{\Pi}^{(q)}$ and $\boldsymbol{\Sigma}^{(q)}$ represent the scaling and permutation ambiguities, respectively, which cannot be removed by ICA. The work [8] assumed $\boldsymbol{\Sigma}^{(q)} = \boldsymbol{I}$ by imposing a unit-variance assumption on all the $z_i^{(q)}$'s. Then, a cross-domain matching algorithm is used to match the shared elements in $\widehat{\boldsymbol{z}}^{(1)}$ and $\widehat{\boldsymbol{z}}^{(2)}$. The formulation can be summarized as finding $d_{\mathrm{C}}$ pairs of non-repetitive $(i, j)$ such that $\boldsymbol{e}_i^\top \widehat{\boldsymbol{z}}^{(1)}$ and $\boldsymbol{e}_j^\top \widehat{\boldsymbol{z}}^{(2)}$ have identical distributions, where $\boldsymbol{e}_i$ is the $i$th unit vector. Denote $\widehat{c}_m^{(1)} = \boldsymbol{e}_{i_m}^\top \widehat{\boldsymbol{z}}^{(1)}$ and $\widehat{c}_m^{(2)} = \boldsymbol{e}_{j_m}^\top \widehat{\boldsymbol{z}}^{(2)}$ for $m \in [d_{\mathrm{C}}]$. It can be shown that

$$\widehat{c}_m^{(q)} = k c_{\boldsymbol{\pi}(m)}^{(q)}, \ m \in [d_{\mathrm{C}}], \tag{5}$$

where $k \in \{+1, -1\}$ and $\boldsymbol{\pi}$ is a permutation of $\{1, \ldots, d_{\mathrm{C}}\}$ (see details in Appendix E.2 summarized from [8]). This method effectively applies ICA to each modality, and thus the ICA identifiability conditions [33] have to met by $\boldsymbol{x}^{(1)}$ and $\boldsymbol{x}^{(2)}$ individually. However, if one only aims to extract $\boldsymbol{\Theta}\boldsymbol{c}$ as in CCA, these assumptions appear to be overly stringent.

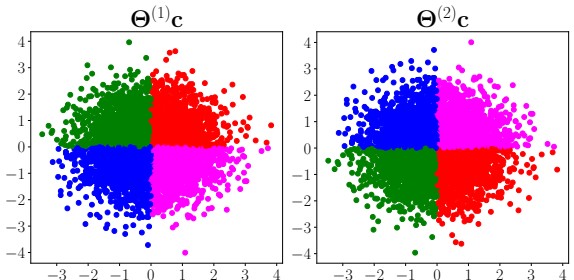 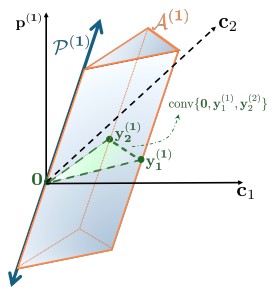

Figure 1: Scatter plots of matched distribution $\boldsymbol{\Theta}^{(1)}\boldsymbol{c}$ (left) and $\boldsymbol{\Theta}^{(2)}\boldsymbol{c}$ (right) when $\boldsymbol{c}$ follows the Gaussian distribution. Colors in the scatter plot represent alignment; same color represent the data are aligned.

Figure 2: Illustration of $\mathcal{A}^{(1)}$ in Assumption 1 in a case where $d_C = 2$ and $d_P^{(1)} = 1$.

## 3 Proposed Approach

**Unaligned SCA: Problem Formulation** We assume that $\boldsymbol{x}^{(q)}$'s are zero-mean. We use the notation from CCA in (3a). However, since no aligned samples are available, we replace the sample-level matching objective with a distribution matching (DM) module, as DM can be carried out without sample level alignment:

$$\text{find} \quad \boldsymbol{Q}^{(q)} \in \mathbb{R}^{d_C \times d^{(q)}}, \ q = 1, 2, \tag{6a}$$

$$\text{subject to} \quad \boldsymbol{Q}^{(1)}\boldsymbol{x}^{(1)} \stackrel{\text{(d)}}{=\!=\!=} \boldsymbol{Q}^{(2)}\boldsymbol{x}^{(2)}, \tag{6b}$$

$$\boldsymbol{Q}^{(q)}\mathbb{E}\left[\boldsymbol{x}^{(q)}(\boldsymbol{x}^{(q)})^\top\right](\boldsymbol{Q}^{(q)})^\top = \boldsymbol{I} \quad q = 1, 2. \tag{6c}$$

where "$\boldsymbol{u} \stackrel{\text{(d)}}{=\!=\!=} \boldsymbol{v}$" means the distributions of $\boldsymbol{u}$ and $\boldsymbol{v}$ are the same.

The formulation in (6) can be realized using various distribution matching tools, e.g., *maximum mean discrepancy* (MMD) [38] and *Wasserstein distance* [39]. We use the adversarial loss:

$$\min_{\boldsymbol{Q}^{(1)}, \boldsymbol{Q}^{(2)}} \max_{f} \ \mathbb{E}_{\boldsymbol{x}^{(1)}} \log\left(f(\boldsymbol{Q}^{(1)}\boldsymbol{x}^{(1)})\right) + \mathbb{E}_{\boldsymbol{x}^{(2)}} \log\left(1 - f(\boldsymbol{Q}^{(2)}\boldsymbol{x}^{(2)})\right) + \lambda \sum_{q=1}^{2} \mathcal{R}\left(\boldsymbol{Q}^{(q)}\right), \tag{7}$$

The first and second terms comprise the adversarial loss from GAN [40]. It finds $\boldsymbol{Q}^{(q)}$ to confuse the best-possible discriminator $f : \mathbb{R}^{d_C} \to \mathbb{R}$, where $f$ is represented by a neural network in practice. It is well known that the minimax optimal point of the first two terms is attained when (6b) is met [40]. We use $\mathcal{R}(\boldsymbol{Q}^{(q)}) = \|\boldsymbol{Q}^{(q)}\mathbb{E}[\boldsymbol{x}^{(q)}(\boldsymbol{x}^{(q)})^\top](\boldsymbol{Q}^{(q)})^\top - \boldsymbol{I}\|_F^2$ to "lift" the constraints. This way, the learning criterion in (7) can be readily handled by any off-the-shelf adversarial learning tools.

**Identifiability of Unaligned SCA** As we saw in Theorem 4, CCA identifies $\widehat{\boldsymbol{Q}}^{(q)}\boldsymbol{x}^{(q)} = \boldsymbol{\Theta}\boldsymbol{c}$ where $\boldsymbol{\Theta} \in \mathbb{R}^{d_C \times d_C}$ under the settings of aligned SCA. Establishing a similar result for unaligned SCA is much more challenging. First, it is unclear if (6b) could disentangle $\boldsymbol{c}$ from $\boldsymbol{p}^{(q)}$. In general, $\boldsymbol{Q}^{(q)}\boldsymbol{x}^{(q)}$ could still be a mixture of $\boldsymbol{c}$ and $\boldsymbol{p}^{(q)}$ yet (6b) still holds (e.g., when both $\boldsymbol{c}$ and $\boldsymbol{p}^{(q)}$ are Gaussian.)

Second, even when the disentanglement is attained via enforcing (6b) and we have $\boldsymbol{Q}^{(q)}\boldsymbol{x}^{(q)} = \boldsymbol{\Theta}^{(q)}\boldsymbol{c}$, in general it does not hold that $\boldsymbol{\Theta}^{(1)} = \boldsymbol{\Theta}^{(2)}$. This is because $\boldsymbol{\Theta}^{(1)}\boldsymbol{c} \stackrel{\text{(d)}}{=\!=\!=} \boldsymbol{\Theta}^{(2)}\boldsymbol{c}$ where $\boldsymbol{\Theta}^{(1)} \neq \boldsymbol{\Theta}^{(2)}$ can still be perfectly met (e.g., when $\mathbb{P}_{\boldsymbol{\Theta}^{(q)}\boldsymbol{c}}$ is symmetric Gaussian in Fig. 1 ). However, $\boldsymbol{\Theta}^{(1)} \neq \boldsymbol{\Theta}^{(2)}$ means that the extracted representations from the two modalities are not matched. This creates challenges for applications like cross-domain information retrieval, language translation, or domain adaptation.

Our intuition is as follows: If the two distributions $\mathbb{P}_{\boldsymbol{c},\boldsymbol{p}^{(1)}}$ and $\mathbb{P}_{\boldsymbol{c},\boldsymbol{p}^{(2)}}$ are very different, then $\boldsymbol{Q}^{(1)}\boldsymbol{x}^{(1)} \stackrel{\text{(d)}}{=\!=\!=} \boldsymbol{Q}^{(2)}\boldsymbol{x}^{(2)}$ cannot hold unless $\boldsymbol{Q}^{(q)}\boldsymbol{A}^{(q)} = [\boldsymbol{\Theta}^{(q)}, \boldsymbol{0}]$. We use the following to characterize such difference between the joint distributions:

**Assumption 1** (Modality Variability). *For any two linear subspaces $\mathcal{P}^{(q)} \subset \mathbb{R}^{d_C + d_P^{(q)}}$, $q = 1, 2$, with $\dim(\mathcal{P}^{(q)}) = d_P^{(q)}$, $\mathcal{P}^{(q)} \neq \mathbf{0} \times \mathbb{R}^{d_P^{(q)}}$ and linearly independent vectors $\{y_i^{(q)} \in \mathbb{R}^{d_C + d_P^{(q)}}\}_{i=1}^{d_C}$, $q = 1, 2$, the sets $\mathcal{A}^{(q)} = \mathrm{conv}\{\mathbf{0}, y_1^{(q)}, \ldots, y_{d_C}^{(q)}\} + \mathcal{P}^{(q)}$, $q = 1, 2$, are such that if $\mathbb{P}_{c, p^{(q)}}[\mathcal{A}^{(q)}] > 0$ for $q = 1$ or $q = 2$, then there exists a $k \in \mathbb{R}$ such that the joint distributions $\mathbb{P}_{c, p^{(1)}}[k\mathcal{A}^{(1)}] \neq \mathbb{P}_{c, p^{(2)}}[k\mathcal{A}^{(2)}]$, where $k\mathcal{A}^{(q)} = \{ka \mid a \in \mathcal{A}^{(q)}\}$.*

The condition in Assumption 1 is a geometric way to characterize the difference between $\mathbb{P}_{c, p^{(1)}}$ and $\mathbb{P}_{c, p^{(2)}}$—if the joint distributions have different measures for all possible "stripes", each being a direct sum of a subspace and a convex hull (see Fig. 2), then $\mathbb{P}_{c, p^{(1)}}$ and $\mathbb{P}_{c, p^{(2)}}$ must be very different. Note that the difference is contributed by the modality-specific term $p^{(q)}$, and thus we call this condition "modality variability". Modality variability is similar to the "domain variablity" used in [32, 41]—both characterize the discrepancy of the joint probabilities $\mathbb{P}_{c, p^{(1)}}$ and $\mathbb{P}_{c, p^{(2)}}$. However, there are key differences: The domain variability was defined in a unified latent domain over *arbitrary* sets $\mathcal{A}$, which could be stringent. Instead, we use the fact that (6) relies on linear operations to construct $\mathcal{A}^{(q)}$, which makes the condition defined over a much smaller class of sets—thereby largely relaxing the requirements. Restricting $\mathcal{A}^{(q)}$ to be stripes also makes the modality variability condition much more relaxed compared to the domain variability condition.

We show the following:

**Theorem 1.** *Under Assumption 1 and the generative model in* (1)*, denote any solution of* (6) *as $\widehat{Q}^{(q)}$ $q = 1, 2$. Then, if the mixing matrices $A^{(q)}$ are full column ranks and $\mathbb{E}[cc^\top])$ is full rank, we have $\widehat{Q}^{(q)} x^{(q)} = \Theta^{(q)} c$. In addition, assume that either of the following is satisfied:*

*(a) The individual elements of the content components are statistically independent and non-Gaussian. In addition, $c_i \overset{(d)}{\neq} kc_j, \forall i \neq j, \forall k \in \mathbb{R}$ and $c_i \overset{(d)}{\neq} -c_i, \forall i$, i.e., the marginal distributions of the content elements cannot be matched with each other by mere scaling.*

*(b) The support of $\mathbb{P}_c$, denoted by $\mathcal{C}$, is a hyper-rectangle, i.e., $\mathcal{C} = [-a_1, a_1] \times \cdots \times [-a_{d_C}, a_{d_C}]$. Further, suppose that $c_i \overset{(d)}{\neq} kc_j, \forall i \neq j, \forall k \in \mathbb{R}$ and $c_i \overset{(d)}{\neq} -c_i, \forall i$.*

*Then, we have $\widehat{Q}^{(q)} x^{(q)} = \Theta c$, i.e., $\Theta^{(q)} = \Theta$ for all $q = 1, 2$, where $\Theta^{(q)}$.*

In Theorem 1, Assumption 1 is used to guarantee $\widehat{Q}^{(q)} x^{(q)} = \Theta^{(q)} c$ and either of conditions (a) or (b) is used to make sure $\Theta^{(1)} = \Theta^{(2)}$. Note that both (a) and (b) are milder than those in [8] (cf. Theorem 5), where the element-wise statistical independence of $z^{(q)}$ was relied on to find shared representation of $x^{(1)}$ and $x^{(2)}$. The proof is in Appendix B.

**Numerical Validation.** In Fig. 3, the top and bottom rows validate Theorem 1 under the assumptions in (a) and (b), respectively. In the top row, we set $c \in \mathbb{R}^2$, where $c_1$ is sampled from Gaussian mixtures with three components and $c_2$ is sampled from a Gamma distribution (and $c_1 \perp\!\!\!\perp c_2$). We set $p^{(1)}$ and $p^{(2)}$ as one-dimensional Laplacian and uniform distributions. In the bottom row, the dimensions of $c$ and $p^{(q)}$ for $q = 1, 2$ are unchanged, but their distributions are replaced in order to satisfy conditions in (b) (see details in Appendix F). One can see that clearly $\widehat{c}^{(q)} = \Theta c$; i.e., the learned $\widehat{c}^{(q)}$ for $q = 1, 2$ are identically rotated and scaled versions of $c$.

A remark is that our framework still allows to identify individual $c_i$'s as in [8].

**Corollary 1.** *Under the conditions in Theorem 1 (a), Assume that at most one $c_i$ for $i \in [d_C]$ is Gaussian. Then, the components of $c$ are identifiable up to permutation and scaling ambiguities by applying ICA to $\widehat{c}^{(q)} = \widehat{Q}^{(q)} x^{(q)}$ for either $q = 1$ or $q = 2$.*

The corollary means that to identify individual $c_i$, using our formulation still enjoys much milder conditions relative to [8]. Specifically, our condition only specifies the independence among elements of $c$, but the condition in [8] needs that all the elements in $z^{(q)} = [c^\top, (p^{(q)})^\top]^\top$ are independent.

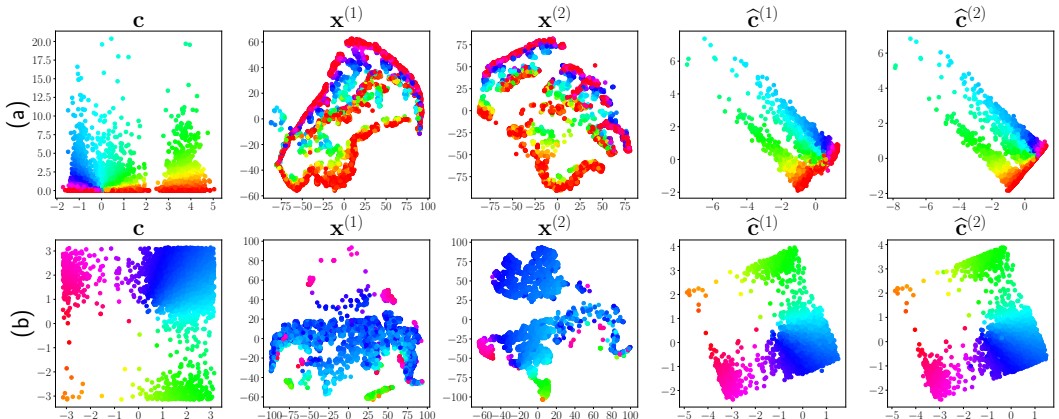

Figure 3: Validation of Theorem 1. Top row: results under assumption (a). Bottom row: results under assumption (b).

## 4 Enhanced Identifiability via Structural Constraints

Theorem 1 was well-supported by the synthetic data experiments. However, our experiments found that the learning criterion (6) often struggles to produce sensible results in some applications. Our conjecture is that the Assumptions in Theorem 1 (a) and (b) might not have been satisfied by the real data under our tests. Although they are not necessary conditions for identifiability, these conditions do indicate that the requirements to guarantee identifiability of unaligned SCA using (6) are nontrivial to meet. In this section, we explore a couple of structural constraints arising from side information in applications to remove the need for the relatively stringent assumptions on $\boldsymbol{c}$.

**Homogeneous Domains.** The first structural constraint that we consider is $\boldsymbol{A}^{(q)} = \boldsymbol{A}$ for $q = 1, 2$. This model is motivated by the fact that advanced representation learning tools, e.g., self-supervised learning tools (e.g., SimCLR [42]) and foundation models (e.g., CLIP [35]), are already capable of mapping the data clusters to a shared linearly separable space—which indicates that the representations share a subspace, i.e., $\boldsymbol{x}^{(q)} \approx \boldsymbol{A}\boldsymbol{z}^{(q)}$. Under such circumstances, the proposed model and method can be used to further process the data by discarding the private components in the latent representation.

Here, we consider the special case of generative process in (1) where,

$$\boldsymbol{x}^{(q)} = \boldsymbol{A}[\boldsymbol{c}^{\top}, (\boldsymbol{p}^{(q)})^{\top}]^{\top}. \tag{8}$$

Under this model, we look for the shared components by solving (6) with a single $\boldsymbol{Q} = \boldsymbol{Q}^{(1)} = \boldsymbol{Q}^{(2)}$. We use the following version of the modality variability condition:

**Assumption 2.** *For any linear subspace $\mathcal{P} \subset \mathbb{R}^{d_{\mathrm{C}}+d_{\mathrm{P}}}$, $d_{\mathrm{P}} = d_{\mathrm{P}}^{(1)} = d_{\mathrm{P}}^{(2)}$, with $\dim(\mathcal{P}) = d_{\mathrm{P}}$, $\mathcal{P} \neq \boldsymbol{0} \times \mathbb{R}^{d_{\mathrm{P}}}$ and linearly independent vectors $\{\boldsymbol{y}_i \in \mathbb{R}^{d_{\mathrm{C}}+d_{\mathrm{P}}}\}_{i=1}^{d_{\mathrm{C}}}$, $q = 1, 2$, the sets $\mathcal{A} = \mathrm{conv}\{\boldsymbol{0}, \boldsymbol{y}_1, \ldots, \boldsymbol{y}_{d_{\mathrm{C}}}\} + \mathcal{P}$, $q = 1, 2$. are such that if $\mathbb{P}_{\boldsymbol{c}, \boldsymbol{p}^{(q)}}[\mathcal{A}] > 0$ for $q = 1$ or $q = 2$, then the joint distributions $\mathbb{P}_{\boldsymbol{c}, \boldsymbol{p}^{(1)}}[k\mathcal{A}] \neq \mathbb{P}_{\boldsymbol{c}, \boldsymbol{p}^{(2)}}[k\mathcal{A}]$ for some $k \in \mathbb{R}$.*

**Theorem 2.** *Consider the mixture model in (8). Assume that $\mathrm{rank}(\boldsymbol{A}) = d_{\mathrm{C}} + d_{\mathrm{P}}$ and $\mathrm{rank}(\mathbb{E}[\boldsymbol{c}\boldsymbol{c}^{\top}]) = d_{\mathrm{C}}$, and that Assumption 2 holds. Denote $\widehat{\boldsymbol{Q}}$ as any solution of (6) by constraining $\boldsymbol{Q} = \boldsymbol{Q}^{(1)} = \boldsymbol{Q}^{(2)}$. Then, we have $\widehat{\boldsymbol{Q}}\boldsymbol{x}^{(q)} = \boldsymbol{\Theta}\boldsymbol{c}$.*

One can see that the conditions (a) and (b) in Theorem 1 are completely removed, if the structure $\boldsymbol{A}^{(1)} = \boldsymbol{A}^{(2)}$ is imposed. In fact, the result in Theorem 2 is expected and readily seen from the proof of Theorem 1, as the cause for $\boldsymbol{\Theta}^{(1)} \neq \boldsymbol{\Theta}^{(2)}$ is the use of two different $\boldsymbol{Q}^{(q)}$'s. Nonetheless, this simple variation will prove useful in a series of real-data experiments.

**The Weakly Supervised Case.** Another way to add structural constraints is to use available auxiliary information. For example, some datasets have weak annotations and selected pairs; see, e.g., [43, 44].

**Assumption 3** (Weak Supervision). *There exist a set of available aligned samples $(\boldsymbol{x}_{\ell}^{(1)}, \boldsymbol{x}_{\ell}^{(2)})$ for $\ell \in \mathcal{L}$ such that $\boldsymbol{x}_{\ell}^{(q)} = \boldsymbol{A}^{(q)}\boldsymbol{z}_{\ell}^{(q)}$, $\boldsymbol{z}_{\ell}^{(q)} = [\boldsymbol{c}_{\ell}^{\top}, (\boldsymbol{p}_{\ell}^{(q)})^{\top}]^{\top}$; i.e., $(\boldsymbol{x}_{\ell}^{(1)}, \boldsymbol{x}_{\ell}^{(2)})$ share the same $\boldsymbol{c}_{\ell}$.*

The condition can be added into our formulation in (6) as a constraint, i.e.,

$$\boldsymbol{Q}^{(1)}\boldsymbol{x}_\ell^{(1)} = \boldsymbol{Q}^{(2)}\boldsymbol{x}_\ell^{(2)}, \; \forall \ell \in \mathcal{L}. \tag{9}$$

In the next theorem, we show that the incorporation of aligned samples helps relax conditions (a) and (b) in Theorem 1:

**Theorem 3.** *Assume that Assumption 1 is satisfied, that $|\mathcal{L}| \geq d_{\mathrm{C}}$ paired samples $(\boldsymbol{x}_\ell^{(1)}, \boldsymbol{x}_\ell^{(2)})$ are available, that $\boldsymbol{A}^{(q)}$ for $q = 1, 2$ have full column rank, and that $\mathbb{P}_{\boldsymbol{c}}$ is absolutely continuous. Denote $(\widehat{\boldsymbol{Q}}^{(1)}, \widehat{\boldsymbol{Q}}^{(2)})$ as any optimal solution of (6) under the constraint (9). Then, we have $\widehat{\boldsymbol{Q}}^{(q)}\boldsymbol{x}^{(q)} = \boldsymbol{\Theta}\boldsymbol{c}$.*

The proof and synthetic data validation can be found in Appendices D and F, respectively. Note that to realize (9), one only needs to add a regularization term $\beta \sum_{\ell \in \mathcal{L}} \|\boldsymbol{Q}^{(1)}\boldsymbol{x}_\ell^{(1)} - \boldsymbol{Q}^{(2)}\boldsymbol{x}_\ell^{(2)}\|_2^2$ to the loss in (7), where $\beta \geq 0$ is a tunable parameter. The overall loss is still differentiable and thus can be easily handled by gradient based approaches.

A remark is that our weakly supervised formulation can use as few as $d_{\mathrm{C}}$ pairs of $(\boldsymbol{x}_\ell^{(1)}, \boldsymbol{x}_\ell^{(2)})$ to establish identifiability of shared component. In contrast, CCA requires at least $d_{\mathrm{C}} + d_{\mathrm{P}}^{(1)} + d_{\mathrm{P}}^{(2)}$ pairs to attain the same identifiability (cf. Appendix. E.1).

**Private Component Identifiability.** Although our focus is shared component identification, we show that private components are also identifiable with additional assumptions; see Appendix H.

## 5 Related Works

*Identifiability of Component Analysis under Linear Mixture Models.* Various component analysis models were studied in the past several decades, e.g., principal component analysis [45], independent component analysis [33], sparse component analysis [10, 12], bounded component analysis [13], simplex component analysis [46, 47], and polytopic component analysis [14]—motivated by their applications in dimensionality reduction, representation learning, and latent variable identification (see, e.g., topic mining [48, 49], hyperspectral unmixing [46, 47], audio/speech separation [33] and community detection [50]). The classical component analysis tools mostly study a single modality. The identifiability results under these models are well developed and documented.

*Identifiability of Shared Components from Aligned Modalities.* Modeling multimodal data as two or more linear/nonlinear mixtures of latent components was considered in CCA-related works [1, 2, 15, 19], *independent vector analysis* (IVA) works [17, 18], multiview ICA works [16, 51], and SSL works [5–7, 52]. Partitioning the latent components into shared and private blocks was considered in [1, 2, 4, 5, 7, 52]. Shared component identifiability was established at the block level (see, e.g., [1, 2, 5]) and the individual component level (e.g., [51]) in these works. Nonetheless, they all rely on completely paired/aligned cross-modality samples, which we do not use in this work.

*Distribution Matching and Unaligned Multimodal Analysis.* Using distribution matching in unaligned multimodal data analytics for different purpose also has a long history; see applications in image-to-image translation [53], domain adaptation [54], cross-platform image super-resolution [55], and cross-domain information retrieval [21]. The recent works [56] and [57] pointed out the identifiability challenge and the existence of density-preserving transforms. The works in [28, 29] started studying the uniqueness issues in distribution matching. However, the latent mixture models were not studied in this line of work.

*Identifiability of Unaligned SCA.* The works in [32, 41] investigated the shared component identifiability when the multimodal data are nonlinear mixtures of content and style (which are shared and private components, respectively) under the same mixing system. Hence, our identical linear mixing case in Theorem 2 can be understood as a special case of theirs. But their analysis relies on the assumption that all the latent components are statistically independent, which is much stronger than our conditions in Theorem 2. Their results also require that there are a large amount of modalities available. But our proof works for just two modalities. The most related work is [8], which uses the model in (1) in the context of multi-view causal graph learning. As discussed before, their assumptions on the latent components are much stronger than ours (see Corollary 1 and Appendix E.2).

Table 1: Classification accuracy on the target domain of *office-31* dataset (ResNet50 embedding).

| source → target | ResNet | DANN | MDD | MCC | SDAT | ELS | Proposed |
|---|---|---|---|---|---|---|---|
| **A → W** | $85.2 \pm 0.2$ | $86.3 \pm 0.3$ | $86.4 \pm 0.4$ | $88.3 \pm 0.3$ | $88.6 \pm 0.4$ | $87.2 \pm 0.3$ | $\mathbf{90.4} \pm 0.4$ |
| **D → W** | $97.5 \pm 0.1$ | $97.4 \pm 0.3$ | $97.7 \pm 0.1$ | $96.9 \pm 0.1$ | $97.6 \pm 0.1$ | $97.7 \pm 0.1$ | $\mathbf{97.8} \pm 0.2$ |
| **W → D** | $99.5 \pm 0.3$ | $98.7 \pm 0.2$ | $\mathbf{99.7} \pm 0.1$ | $97.4 \pm 0.2$ | $99.1 \pm 0.2$ | $99.3 \pm 0.2$ | $99.5 \pm 0.3$ |
| **A → D** | $89.4 \pm 0.2$ | $84.3 \pm 0.4$ | $89.9 \pm 0.2$ | $87.4 \pm 0.5$ | $86.3 \pm 0.4$ | $87.1 \pm 0.2$ | $\mathbf{90.1} \pm 0.3$ |
| **D → A** | $71.4 \pm 0.3$ | $71.7 \pm 0.4$ | $70.6 \pm 0.3$ | $\mathbf{74.9} \pm 0.4$ | $72.3 \pm 0.4$ | $71.6 \pm 0.3$ | $71.9 \pm 0.1$ |
| **W → A** | $73.1 \pm 0.2$ | $73.5 \pm 0.2$ | $72.3 \pm 0.4$ | $73.0 \pm 0.4$ | $73.6 \pm 0.3$ | $73.7 \pm 0.3$ | $\mathbf{74.6} \pm 0.1$ |
| **Average** | $86.0 \pm 0.2$ | $85.3 \pm 0.3$ | $86.1 \pm 0.2$ | $86.3 \pm 0.3$ | $86.2 \pm 0.3$ | $86.1 \pm 0.2$ | $\mathbf{87.3} \pm 0.2$ |

## 6 Numerical Validation

**More Synthetic-Data Validation.** We first validate our proposed method on synthetic data that follows our model; see Appendix F for details.

**Application (i) - Domain Adaptation.** We first test the proposed methods over a number of domain adaptation (DA) tasks. In DA, we have the source domain data $\{x^{(1)}\}$ and the target domain $\{x^{(2)}\}$, respectively. Only the source domain data have labels and the two domains are unaligned. We hope to use our method to find shared representations of source and target, and thus the classifier trained using source data can also work well on the target data.

*Dataset*: We use two standard benchmarks of DA, i.e., *Office-31* [58] and *Office-Home* [59]. The *Office-31* dataset has 4652 images and 31 categories from three domains, namely, Amazon images (**A**), Webcam images (**W**) and DSLR images (**D**). The *Office-Home* dataset contains 15,500 images with 65 object classes from four domains, i.e., Artistic images (**Ar**), Clip art images (**Cl**), Product images (**Pr**), and Real-world images (**Rw**).

*Setup*: We first test the homogeneous domain model in Sec. 4. The images are pre-processed using a ResNet50-based image encoder pre-trained over ImageNet1k [42]. As mentioned, it was observed that self-supervised representation encoders find embeddings that are linearly separable [42], which justifies the use of the model $x^{(q)} \approx A z^{(q)}$ in the embedding domain. After pre-processing, each image is represented by $d^{(q)} = 2048$ features for $q = 1, 2$. We set $d_{\mathrm{C}} = 256$ for *Office-31* and $d_{\mathrm{C}} = 512$ for *Office-Home*. More detailed settings are in Appendix G.

*Baselines and Training Setup*: The baselines are representative DA methods, namely, DANN [25], MDD [60], MCC [61], SDAT [62], and ELS [63]. All the baselines use the same encoder-produced embeddings as inputs; see Appendix G.1 for their configurations. We also use ResNet encoder's outputs as an extra baseline as it learns informative and transferable features from the ImageNet-1K dataset. We follow the training strategies adopted by the baselines [25, 60, 62] to learn a classifier jointly with the shared latent components. This strategy arguably regularizes towards more classification-friendly geometry of the shared features. Therefore we append a cross-entropy (CE) based classifier training module to our loss in (7) that learns our feature extractor $Q$. More details are in Appendix G.1.

*Metric*: The evaluation metric is the classification accuracy in the target domain $\{x^{(2)}\}$. The classifier is trained with the projected source domain $\widehat{Q} x^{(1)}$ and the associated labels.

*Result*: Table 1 and Table 2 show the classification accuracy (mean±std) on *Office-31* and *Office-Home*, respectively. The results are averaged over 5 runs. One can observe that the proposed method offers the best and second best performance in most of the cases. In some tasks (e.g.,"**A→W**", "**Ar→Cl**", "**Ar→Pr**" and "**Rw→Cl**"), the proposed method outperforms the best-performing baselines by at least 2% in accuracy.

More results on the DA task can be found in Appendix G.1.

**Application (ii) - Single Cell Sequence Analysis.** In biomedical research, it is desired to fuse measurements from multiple sensorial modalities of the same cells, in order to have better characterizations of the cells. However, obtaining multimodal data of the same cells simultaneously is almost impossible, due to the sensing limitations. Therefore, many methods are proposed in the literature for aligning unpaired multi-modal single cell data [27, 64, 65]. We focus on the following two modalities of single-cell data [66]: (1) the RNA sequences $\{x^{(1)}\}$ and (2) the ATAC sequences $\{x^{(2)}\}$.

Table 2: Classification accuracy on the target domain of *office-Home* dataset (ResNet50 embedding).

| source → target | ResNet | DANN | MDD | MCC | SDAT | ELS | Proposed |
|---|---|---|---|---|---|---|---|
| **Ar → Cl** | $42.0 \pm 0.2$ | $46.7 \pm 0.2$ | $47.4 \pm 0.3$ | $44.4 \pm 0.3$ | $47.3 \pm 0.4$ | $48.5 \pm 0.2$ | $\mathbf{51.0 \pm 0.3}$ |
| **Ar → Pr** | $69.2 \pm 0.1$ | $70.2 \pm 0.4$ | $72.8 \pm 0.4$ | $72.4 \pm 0.2$ | $71.1 \pm 0.3$ | $71.0 \pm 0.3$ | $\mathbf{75.8 \pm 0.1}$ |
| **Ar → Rw** | $80.2 \pm 0.3$ | $81.2 \pm 0.4$ | $81.2 \pm 0.1$ | $80.3 \pm 0.3$ | $80.5 \pm 0.1$ | $80.8 \pm 0.4$ | $\mathbf{82.5 \pm 0.2}$ |
| **Cl → Ar** | $60.7 \pm 0.4$ | $60.8 \pm 0.3$ | $62.4 \pm 0.1$ | $59.2 \pm 0.4$ | $57.6 \pm 0.2$ | $59.8 \pm 0.1$ | $\mathbf{62.7 \pm 0.4}$ |
| **Cl → Pr** | $71.0 \pm 0.1$ | $69.8 \pm 0.3$ | $70.0 \pm 0.4$ | $\mathbf{71.1 \pm 0.4}$ | $66.5 \pm 0.1$ | $68.5 \pm 0.2$ | $72.5 \pm 0.3$ |
| **Cl → Rw** | $74.8 \pm 0.2$ | $73.3 \pm 0.1$ | $74.1 \pm 0.1$ | $\mathbf{76.2 \pm 0.2}$ | $70.7 \pm 0.1$ | $71.7 \pm 0.1$ | $75.8 \pm 0.1$ |
| **Pr → Ar** | $60.6 \pm 0.2$ | $62.2 \pm 0.1$ | $64.3 \pm 0.1$ | $59.2 \pm 0.1$ | $62.5 \pm 0.4$ | $60.9 \pm 0.2$ | $\mathbf{64.4 \pm 0.3}$ |
| **Pr → Cl** | $44.8 \pm 0.1$ | $48.8 \pm 0.1$ | $48.0 \pm 0.3$ | $46.2 \pm 0.2$ | $49.0 \pm 0.3$ | $49.6 \pm 0.3$ | $\mathbf{50.4 \pm 0.1}$ |
| **Pr → Rw** | $79.6 \pm 0.1$ | $80.3 \pm 0.4$ | $79.6 \pm 0.3$ | $80.3 \pm 0.2$ | $80.0 \pm 0.1$ | $79.2 \pm 0.1$ | $\mathbf{81.7 \pm 0.2}$ |
| **Rw → Ar** | $70.1 \pm 0.2$ | $71.5 \pm 0.1$ | $71.4 \pm 0.3$ | $67.8 \pm 0.2$ | $71.6 \pm 0.4$ | $71.3 \pm 0.4$ | $\mathbf{72.6 \pm 0.1}$ |
| **Rw → Cl** | $45.8 \pm 0.2$ | $50.9 \pm 0.2$ | $50.3 \pm 0.1$ | $50.0 \pm 0.2$ | $51.4 \pm 0.1$ | $50.7 \pm 0.1$ | $\mathbf{53.2 \pm 0.1}$ |
| **Rw → Pr** | $80.7 \pm 0.1$ | $80.6 \pm 0.4$ | $81.1 \pm 0.1$ | $81.2 \pm 0.1$ | $80.7 \pm 0.1$ | $79.8 \pm 0.3$ | $\mathbf{82.9 \pm 0.3}$ |
| **Average** | $64.9 \pm 0.1$ | $66.3 \pm 0.2$ | $66.8 \pm 0.2$ | $65.6 \pm 0.2$ | $65.7 \pm 0.2$ | $65.9 \pm 0.2$ | $\mathbf{68.7 \pm 0.2}$ |

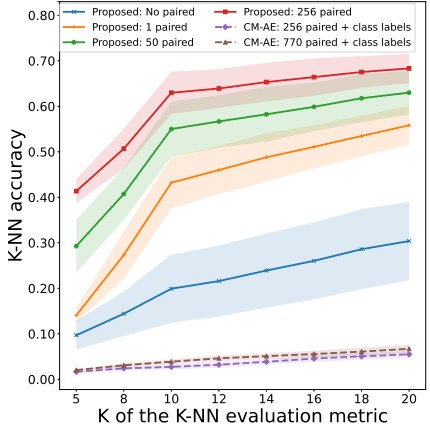

Figure 4: $k$-NN accuracy for single-cell sequence alignment.

Table 3: Average precision P@1 of cross-language information retrieval.

| source → target | Adv - NN | proposed - NN | Adv - CSLS | proposed - CSLS |
|---|---|---|---|---|
| **en→es** | 61.3 | **66.4** | 70.2 | **74.9** |
| **es→en** | 55.4 | **65.3** | 67.6 | **75.6** |
| **en→it** | 48.2 | **54.4** | 60.8 | **67.7** |
| **it→en** | **55.2** | 51.9 | 63.8 | **66.0** |
| **en→fr** | **63.6** | 60.2 | 72.6 | **73.7** |
| **fr→en** | 55.4 | **58.4** | 64.1 | **71.4** |
| **en→de** | 51.4 | **56.7** | 59.3 | **67.6** |
| **de→en** | 42.5 | **57.0** | 51.0 | **59.3** |
| **en→ru** | 32.7 | **34.9** | 38.6 | **41.4** |
| **ru→en** | 27.6 | **41.6** | 35.0 | **50.8** |
| **en→ar** | 12.6 | **22.7** | 16.7 | **29.1** |
| **ar→en** | 15.7 | **26.9** | 20.1 | **35.6** |
| **en→vi** | 2.1 | **10.4** | 7.7 | **22.8** |
| **vi→en** | 2.7 | **17.3** | 4.4 | **33.0** |
| **Average** | 37.6 | **44.5** | 45.1 | **54.9** |

*Dataset*: We use human lung adenocarcinoma A549 cells data from [66]. The dataset contains 1,874 samples of RNA sequences $\{x^{(1)}\}$ and ATAC sequences $\{x^{(2)}\}$. Each data set is split into 1534 training samples and 340 testing samples as in [27]. The data have labeled associations between the two domains—part of which will be used to test our weakly supervised formulation. For this experiment, features of RNA sequence and the ATAC sequence have dimensions of $d^{(1)} = 815$ and $d^{(2)} = 2613$, respectively. We set $d_C = 256$. We use our weakly supervised formulation as shown in (9). We uniformly sampled a set of indices from the training set to serve as $\mathcal{L}$.

*Baseline and Metric*: We use weakly supervised algorithm, namely, cross-modal autoencoder (CM-AE) work in [27], as a baseline, which also learns the shared representation between unaligned RNA and ATAC sequences. We use the $K$-nearest neighbor ($k$-NN) accuracy to evaluate the performance as suggested in [27].

*Result*: The plot in Fig. 4 shows the $k$-NN accuracy of the methods on the test set. Results show the mean and standard deviation over 10 runs, each having a different random initialization. For the proposed method, we vary the number of available paired samples from 0 (cf. Theorem 1) to $d_C = 256$ (cf. Theorem 3). Note that the baseline uses more (i.e., 256 and 770) paired samples. It also needs additional class labels, i.e., $y_i^{(q)}$ for the $i$th sample $x_i^{(q)}$. Here, $y_i^{(q)}$ represents the number of hours (0, 1 or 3) of cell treatment [27, 66]. The proposed method without any supervision (i.e., 0 paired samples) already exhibits around 3 times greater $k$-NN accuracy compared to the baseline for all $k$. Moreover, including just one paired sample boosts the $k$-NN accuracy of the proposed method to around 5 times higher than the baseline for all $k$. Finally, one can observe a steadily increasing $k$-NN accuracy with respect to the number of available paired samples. This corroborates with our Theorem 3.

**Application (iii) - Multi-lingual Information Retrieval.** We also evaluated our method on a word embedding association problem from the natural language processing literature [20, 21]. This task aims to associate high-dimensional word embeddings across different languages according to their semantic meaning. The word embeddings in two languages are represented using two sets of vectors, i.e., $\{\boldsymbol{x}_i^{(1)}\}_{i=1}^I$ and $\{\boldsymbol{x}_j^{(2)}\}_{j=1}^J$. The postulate is that if $\boldsymbol{x}_i^{(1)}$ and $\boldsymbol{x}_j^{(2)}$ have the same meaning (e.g., both representing "cat") in two languages (e.g., English and German), they should share a latent components $\boldsymbol{c}$.

*Dataset*: We use the word embeddings from the MUSE dataset (`https://github.com/facebookresearch/MUSE`) [21]. These monolingual word embedding are generated using fast-Text [67] and has dimensions of $d^{(q)} = 300$ for $q = 1, 2$. The training dataset include 200,000 word embeddings in each language. In our experiment we set $d_{\mathrm{C}} = 256$. We follow the generative model under (8) and run the formulation in (7) to learn the linear transformation $\boldsymbol{Q}$.

*Baseline*: We use `Adv` [21] as the baseline which also uses distribution matching between two language domains. Unlike our method, `Adv` does not use linear mixture models.

*Metric*: We follow [21] to use the average precision score calculated based on *nearest neighbor* (NN) and *cross domain similarity local scaling* (CSLS). Precision at $k$ ("$k$ precision") is computed by the number of times that one of the correct translations of source word is retrieved at top-$k$ results ($k = \{1, 5, 10\}$). The final score is normalized to be in the range of 0 to 100, with 100 being the highest score indicating the best performance. To evaluate the performance, we use the same test data as in [21]. For each source and target language pair, this dataset includes 1,500 source word embeddings. The source embeddings are used to retrieve corresponding embeddings from a pool of 200,000 target word embeddings.

*Result*: Table 3 reports the P@1 scores over the test data calculated for each source and target language pair. The languages are denoted as as **en** - English, **es** - Spanish, **it** - Italian, **fr** - French, **de** - Germany, **ru** - Russian, **ar** - Arabic and **vi** - Vietnamese. One can observe that the proposed method exhibits a better precision performance than that of `Adv` in most of the translation tasks. In particular, the proposed method significantly outperforms the baseline on the tasks **en→ar**, **ar→en**, **en→vi** and **vi→en**, showing at least $10\%$ precision gains. Similarly, our method shows at least $5\%$ improvements in both NN and CSLS based precision metrics in **en→es** and **es→en** tasks.

More details and additional experiments can be found in Appendix G.3.

## 7 Conclusion

In this work, we considered the problem of identifying shared components from unaligned multi-domain mixtures. We proposed a learning loss that matches the distributions of linearly transformed data. Based on this loss, we came up with a suite of sufficient conditions to ensure the identifiability of shared components. Furthermore, we proposed modified models and losses that enjoy more relaxed conditions for shared component identifiability. This was achieved via introducing structural constraints, namely, the homogeneity of the mixing systems and the existence of weak supervision. Our theoretical claims were validated with both synthetic and real-world data, demonstrating soundness of the theorems and usefulness of the models/algorithms.

**Limitations.** First, our conditions for shared component identification are sufficient. The necessary conditions are not underpinned, but necessary conditions assist understanding the limitations of the models and algorithms. Second, our methods were developed under the linear mixture model, which has limited expressiveness, and thus often requires pre-processing to approximately meet the model specification. We expect that results with similar flavors to be derived for nonlinear models in the future. Third, the results were derived under an unlimited data assumption. It would be interesting have a finite sample analysis. Finally, optimizing GAN-based losses is sensitive to hyperparameter settings. Back-propagation based minimax optimization occasionally fails to converge. More optimization-friendly losses and more stable algorithms are desirable in the context of distribution matching.

## Acknowledgment

This work is supported in part by the Army Research Office (ARO) under Project ARO W911NF-21-1-0227, and in part by the National Science Foundation (NSF) CAREER Award ECCS-2144889.

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

# Supplementary Material of "Identifiable Shared Component Analysis of Unpaired Multimodal Mixtures"

## A  Notation

The notations used throughout the paper are summarized in the Table 4.:

Table 4: Definition of notations.

| Notation | Definition |
|---|---|
| $x, \boldsymbol{x}, \boldsymbol{X}$ | scalar, vector and matrix |
| $\boldsymbol{x}^{(q)}$ | variable from q-th domain |
| $x_i, \boldsymbol{x}_i$ | both represents i-th element of vector $\boldsymbol{x}$ |
| $\boldsymbol{X}_{ij}$ | represents the element of i-th row and j-th column of matrix $\boldsymbol{X}$ |
| $\boldsymbol{x}^\top, \boldsymbol{X}^\top$ | transpose of $\boldsymbol{x}$ and $\boldsymbol{X}$ |
| $|\mathcal{X}|$ | represents the cardinality of set $\mathcal{X}$ |
| $\mathsf{Null}(\boldsymbol{X})$ | represents the null space of matrix $\boldsymbol{X}$ |
| $\mathrm{conv}(\cdot)$ | returns the convex hull of the given set |
| $\dim(\mathcal{X})$ | denotes the dimension of subspace $\mathcal{X}$ |
| $k\mathcal{A}$ | $\{k\boldsymbol{a} \mid \boldsymbol{a} \in \mathcal{A}, \ k \in \mathbb{R}\}$ |
| $\boldsymbol{x} + \mathcal{X}$ | $\{\boldsymbol{x} + \boldsymbol{z} \mid \boldsymbol{z} \in \mathcal{X}\}$ |
| $\mathcal{X} + \mathcal{Y}$ | $\{\boldsymbol{x} + \boldsymbol{y} \mid \boldsymbol{x} \in \mathcal{X}, \boldsymbol{y} \in \mathcal{Y}\}$ |
| $\boldsymbol{A}_{\mathrm{PreImg}}(\mathcal{X})$ | preimage of $\mathcal{X}$; $\{\boldsymbol{x} \mid \boldsymbol{A}\boldsymbol{x} \in \mathcal{X}\}$ |
| $[N]$ | set of whole numbers up to $N$; $\{1 \ldots N\}$ |
| $\boldsymbol{I}$ | identity matrix |
| $\mathbb{P}_{\boldsymbol{x}}$ | probability distribution of random variable $\boldsymbol{x}$ |
| $\mathbb{P}_{\boldsymbol{x},\boldsymbol{y}}$ | joint probability distribution of random variable $\boldsymbol{x}$ and $\boldsymbol{y}$ |
| $\mathbb{E}[\cdot]$ | expectation |
| $\boldsymbol{x} \overset{(d)}{=\!=} \boldsymbol{y}$ | $\boldsymbol{x}$ and $\boldsymbol{y}$ random vectors have the same distribution |
| $\boldsymbol{x} \overset{(d)}{\neq} \boldsymbol{y}$ | $\boldsymbol{x}$ and $\boldsymbol{y}$ random vectors have different distributions |
| $\boldsymbol{x} \perp\!\!\!\perp \boldsymbol{y}$ | $\boldsymbol{x}$ and $\boldsymbol{y}$ random vectors are statistically independent |
| $[a, b]$ | represents continuous interval between $a$ and $b$ |
| $\mathcal{N}(\mu, \sigma^2)$ | normal distribution with mean $\mu$ and variance $\sigma^2$ |
| $\mathtt{Uniform}[a, b]$ | uniform distribution with interval $a$ and $b$ |
| $\mathtt{Gamma}(\alpha, \ \theta)$ | gamma distribution with the shape parameter $\alpha$ and scale parameter $\theta$ |
| $\mathtt{Laplace}(\mu, \ b)$ | Laplace distribution with location $\mu$ and diversity or scale parameter $b$ |
| $\mathtt{VonMises}(\mu, \kappa)$ | von Mises distribution with location $\mu$ and $\kappa$ concentration parameter. |
| $\mathtt{Beta}(\alpha, \beta)$ | beta distribution with the shape parameters $\alpha$ and $\beta$ |

# B  Proof of Theorem 1

We restate the theorem here:

**Theorem** 1 Under Assumption 1 and the generative model in (1), denote any solution of (6) as $\widehat{\boldsymbol{Q}}^{(q)}$ $q = 1, 2$. Then, if the mixing matrices $\boldsymbol{A}^{(q)}$ are full column ranks and $\mathbb{E}[\boldsymbol{c}\boldsymbol{c}^\top]$ is full rank, we have $\widehat{\boldsymbol{Q}}^{(q)}\boldsymbol{x}^{(q)} = \boldsymbol{\Theta}^{(q)}\boldsymbol{c}$. In addition, assume that either of the following is satisfied:

(a) The individual elements of the content components are statistically independent and non-Gaussian. In addition, $c_i \overset{(d)}{\neq} kc_j, \forall i \neq j, \forall k \in \mathbb{R}$ and $c_i \overset{(d)}{\neq} -c_i, \forall i$, i.e., the marginal distributions of the content elements cannot be matched with each other by mere scaling.

(b) The support $\mathcal{C}$ is a hyper-rectangle, i.e., $\mathcal{C} = [-a_1, a_1] \times \cdots \times [-a_{d_C}, a_{d_C}]$. Further, suppose that $c_i \overset{(d)}{\neq} kc_j, \forall i \neq j, \forall k \in \mathbb{R}$ and $c_i \overset{(d)}{\neq} -c_i, \forall i$.

Then, we have $\widehat{\boldsymbol{Q}}^{(q)}\boldsymbol{x}^{(q)} = \boldsymbol{\Theta}\boldsymbol{c}$, i.e., $\boldsymbol{\Theta}^{(q)} = \boldsymbol{\Theta}$ for all $q = 1, 2$, where $\boldsymbol{\Theta}^{(q)}$.

We will prove the theorem in following two steps. For the first step we will prove $\widehat{\boldsymbol{Q}}^{(q)}\boldsymbol{x}^{(q)} = \boldsymbol{\Theta}^{(q)}\boldsymbol{c}$ and for second step we will employ either assumption (a) or (b) to prove that $\boldsymbol{\Theta}^{(q)} = \boldsymbol{\Theta}$, $\forall q = 1, 2$.

## B.1  Linearly transformed content identification

Let us define

$$\boldsymbol{H}^{(q)} = \boldsymbol{Q}^{(q)}\boldsymbol{A}^{(q)} \in \mathbb{R}^{d_C \times (d_C + d_P^{(q)})}.$$

We want to show that

$$\text{Null}(\boldsymbol{H}^{(q)}) = \mathbf{0} \times \mathbb{R}^{d_P^{(q)}}, \tag{10}$$

since this will imply that that $\boldsymbol{H}^{(q)}$ does not depend upon the style component. Combined with the fact that $\text{rank}(\boldsymbol{H}^{(q)}) = d_C$, this will imply that $\boldsymbol{H}^{(q)}$ is an invertible function of the content component. To that end, consider the following line of arguments.

Since the objective in (6) matches the distribution for latent random variables $\widehat{\boldsymbol{c}}^{(1)} = \boldsymbol{Q}^{(1)}\boldsymbol{x}^{(1)}$ and $\widehat{\boldsymbol{c}}^{(2)} = \boldsymbol{Q}^{(2)}\boldsymbol{x}^{(2)}$, the following holds for any $\mathcal{R}_c \subseteq \mathbb{R}^{d_C}, \forall k \in \mathbb{R}$,

$$\mathbb{P}_{\widehat{\boldsymbol{c}}^{(1)}}[k\mathcal{R}_c] = \mathbb{P}_{\widehat{\boldsymbol{c}}^{(2)}}[k\mathcal{R}_c],$$

$$\overset{(a)}{\Longleftrightarrow} \mathbb{P}_{\boldsymbol{z}^{(1)}}[\boldsymbol{H}_{\text{PreImg}}^{(1)}(k\mathcal{R}_c)] = \mathbb{P}_{\boldsymbol{z}^{(2)}}[\boldsymbol{H}_{\text{PreImg}}^{(2)}(k\mathcal{R}_c)] \tag{11}$$

$$\overset{(b)}{\Longleftrightarrow} \mathbb{P}_{\boldsymbol{z}^{(1)}}[k\boldsymbol{H}_{\text{PreImg}}^{(1)}(\mathcal{R}_c)] = \mathbb{P}_{\boldsymbol{z}^{(2)}}[k\boldsymbol{H}_{\text{PreImg}}^{(2)}(\mathcal{R}_c)],$$

where, $\boldsymbol{H}_{\text{PreImg}}^{(q)}(\mathcal{R}_c) := \{\boldsymbol{z}^{(q)} \mid \boldsymbol{H}^{(q)}\boldsymbol{z}^{(q)} \in \mathcal{R}_c\}$ is the pre-image of $\boldsymbol{H}^{(q)}$. $(a)$ follows because $\mathbb{P}_{\widehat{\boldsymbol{c}}^{(q)}}[k\mathcal{R}_c] = \mathbb{P}_{\boldsymbol{H}^{(q)}\boldsymbol{z}^{(q)}}[k\mathcal{R}_c] = \mathbb{P}_{\boldsymbol{z}^{(q)}}[\boldsymbol{H}_{\text{PreImg}}^{(q)}(k\mathcal{R}_c)]$ [68, Section 2.2]. $(b)$ follows because $\boldsymbol{H}^{(q)}$ is a linear operator.

Although (11) holds for any $\mathcal{R}_c$, we will see that it is sufficient to consider a special $\mathcal{R}_c$ to prove (10). To that end, take $\mathcal{R}_c = \text{conv}\{\mathbf{0}, \boldsymbol{a}_1, \ldots, \boldsymbol{a}_{d_C}\}$, where $\boldsymbol{a}_i \in \mathbb{R}^{d_C}$ such that $\mathbb{P}_{\widehat{\boldsymbol{c}}^{(q)}}[\mathcal{R}_c] > 0$. Let us take $\boldsymbol{y}_i^{(q)} \in \mathbb{R}^{d_C + d_P^{(q)}}$, such that $\boldsymbol{H}^{(q)}\boldsymbol{y}_i^{(q)} = \boldsymbol{a}_i$. For reasons that will be clear later, we hope to show that

$$\boldsymbol{H}_{\text{PreImg}}^{(q)}(\mathcal{R}_c) = \text{conv}\{\mathbf{0}, \boldsymbol{y}_1^{(q)}, \ldots, \boldsymbol{y}_{d_C}^{(q)}\} + \text{Null}(\boldsymbol{H}^{(q)}).$$

To that end, observe that for any $\boldsymbol{r} \in \mathcal{R}_c$, we can represent $\boldsymbol{r}$ as,

$$\boldsymbol{r} = \frac{1}{d_C + 1}\sum_{i=1}^{d_C} w_i \boldsymbol{a}_i, \text{ for some } \{w_i\}_{i=1}^{d_C} \text{ s.t. } \sum_{i=1}^{d_C} w_i \leq 1, \forall i.$$

For both view $q = 1, 2$, we get,

$$r = \frac{1}{d_C + 1} \sum_{i=1}^{d_C} w_i \boldsymbol{H}^{(q)} \boldsymbol{y}_i^{(q)}$$

$$\implies r = \boldsymbol{H}^{(q)} \left( \frac{1}{d_C + 1} \sum_{i=1}^{d_C} w_i \boldsymbol{y}_i^{(q)} \right)$$

$$\boldsymbol{H}_{\text{PreImg}}^{(q)} \left( \frac{1}{d_C + 1} \sum_{i=1}^{d_C} w_i \boldsymbol{a}_i \right) = \frac{1}{d_C + 1} \sum_{i=1}^{d_C} w_i \boldsymbol{y}_i^{(q)} + \text{Null}(\boldsymbol{H}^{(q)}) \tag{12}$$

We can write,

$$\boldsymbol{H}_{\text{PreImg}}^{(q)}(\mathcal{R}_c) = \text{conv}\{\boldsymbol{0}, \boldsymbol{y}_1^{(q)}, \ldots, \boldsymbol{y}_{d_C}^{(q)}\} + \text{Null}(\boldsymbol{H}^{(q)}) \tag{13}$$

We have that $\text{Null}(\boldsymbol{H}^{(q)}) \subset \mathbb{R}^{d_C + d_P^{(q)}}$ is a linear subspace with $\dim(\text{Null}(\boldsymbol{H}^{(q)})) = d_P^{(q)}$. Let $\mathcal{A}^{(q)} = \boldsymbol{H}_{\text{PreImg}}^{(q)}(\mathcal{R}_c)$. Note that $\mathbb{P}_{\boldsymbol{z}^{(1)}}[k\mathcal{A}^{(1)}] = \mathbb{P}_{\boldsymbol{z}^{(2)}}[k\mathcal{A}^{(2)}], \forall k \in \mathbb{R}$ (from (11)), and $\mathbb{P}_{\boldsymbol{z}^{(q)}}[\mathcal{A}^{(q)}] > 0$ (by the construction of $\mathcal{R}_c$). Further, the set $\mathcal{A}^{(q)}$ is of the form

$$\text{conv}\{\boldsymbol{0}, \boldsymbol{y}_1^{(q)}, \ldots, \boldsymbol{y}_{d_C}^{(q)}\} + \mathcal{P}^{(q)},$$

because $\text{Null}(\boldsymbol{H}^{(q)})$ is a subspace of dimension $d_P^{(q)}$, hence it satisfies the definition of $\mathcal{P}^{(q)}$. Hence, Assumption 1 implies that

$$\text{Null}(\boldsymbol{H}^{(q)}) = \boldsymbol{0} \times \mathbb{R}^{d_P^{(q)}}.$$

Denoting the $N$th to $M$th columns of $\boldsymbol{H}^{(q)}$ by $\boldsymbol{H}^{(q)}(N : M)$, the above is equivalent to saying

$$\boldsymbol{H}^{(q)}(d_C + 1 : d_C + d_P^{(q)}) = 0. \tag{14}$$

Denote,

$$\boldsymbol{\Theta}^{(q)} = \boldsymbol{H}^{(q)}(1 : d_C) \,\forall\, q = 1, 2.$$

Then, we can write,

$$\boldsymbol{Q}^{(q)} \boldsymbol{x}^{(q)} = \boldsymbol{\Theta}^{(q)} \boldsymbol{c}, \,\forall\, q = 1, 2. \tag{15}$$

Next, we use Assumption (a) or (b) to show that $\boldsymbol{\Theta}^{(1)} = \boldsymbol{\Theta}^{(2)} = \boldsymbol{\Theta}$. To that end, note that the distribution matching constraint implies that

$$\boldsymbol{\Theta}^{(1)} \boldsymbol{c} \overset{\text{(d)}}{=\!=\!=} \boldsymbol{\Theta}^{(2)} \boldsymbol{c}$$

$$\implies \boldsymbol{c} \overset{\text{(d)}}{=\!=\!=} (\boldsymbol{\Theta}^{(1)})^{-1} \boldsymbol{\Theta}^{(2)} \boldsymbol{c}.$$

Hence $\boldsymbol{M} = (\boldsymbol{\Theta}^{(1)})^{-1} \boldsymbol{\Theta}^{(2)}$ is an invertible matrix such that $\boldsymbol{c} \overset{\text{(d)}}{=\!=\!=} \boldsymbol{M} \boldsymbol{c}$. However, in the following, we will show that if either Assumption (a) or (b) is satisfied, then $\boldsymbol{M} = \boldsymbol{I}$, and thus $\boldsymbol{\Theta}^{(1)} = \boldsymbol{\Theta}^{(2)}$.

## B.2 Considering Assumption (a)

We want to show that when Assumption (a) is satisfied, if $\boldsymbol{M} \boldsymbol{c} \overset{\text{(d)}}{=\!=\!=} \boldsymbol{c}$ for any invertible $\boldsymbol{M}$, then $\boldsymbol{M} = \boldsymbol{I}$.

Note that $\boldsymbol{M} \boldsymbol{c} = [\boldsymbol{m}_1 \ldots \boldsymbol{m}_{d_C}] \begin{bmatrix} c_1 \\ \vdots \\ c_{d_C} \end{bmatrix}$. By Assumption (a), we have that the components of content are statistically independent $c_i \perp\!\!\!\perp c_j$, $i \neq j$, non-Gaussian, and has non-zero kurtosis. Then, according to cumulant multilinearity and additivity properties, the fourth order cumulant tensor $\text{Cum}(\boldsymbol{M} \boldsymbol{c})$ of $\boldsymbol{M} \boldsymbol{c}$ has the following unique decomposition [69],

$$\text{Cum}(\boldsymbol{M} \boldsymbol{c}) = \sum_{i=1}^{d_C} \kappa_i \boldsymbol{m}_i \circ \boldsymbol{m}_i \circ \boldsymbol{m}_i \circ \boldsymbol{m}_i \tag{16}$$

where $\circ$ is the outer product, $\kappa_i$ is the kurtosis of component $c_i$, and $\boldsymbol{m}_i$, $i \in [d_C]$ are the columns of $\boldsymbol{M}$.

Since $\boldsymbol{M}\boldsymbol{c} \overset{(d)}{=\!=} \boldsymbol{c}$, the following should hold

$$\mathrm{Cum}(\boldsymbol{M}\boldsymbol{c}) = \mathrm{Cum}(\boldsymbol{c}) = \mathrm{Cum}(\boldsymbol{I}\boldsymbol{c}) \tag{17}$$

$$\implies \sum_{d=1}^{d_C} \kappa_d \, \boldsymbol{m}_d \circ \boldsymbol{m}_d \circ \boldsymbol{m}_d \circ \boldsymbol{m}_d = \sum_{d=1}^{d_C} \kappa_d \, \boldsymbol{e}_d \circ \boldsymbol{e}_d \circ \boldsymbol{e}_d \circ \boldsymbol{e}_d, \tag{18}$$

$\boldsymbol{e}_i$ is the $i$th column of identity matrix $\boldsymbol{I}$.

Because of statistical independence of components of $\boldsymbol{c}$, the CP-decomposition of $\mathrm{Cum}(\boldsymbol{M}\boldsymbol{c}) = \mathrm{Cum}(\boldsymbol{I}\boldsymbol{c})$ is unique [69] upto permutation and scaling ambiguities, i.e., $\boldsymbol{M}$ should be a permutation scaling matrix.

Let $\boldsymbol{M} = \boldsymbol{\Pi}\boldsymbol{\Sigma}$ where, $\boldsymbol{\Pi} \in \mathbb{R}^{d_C \times d_C}$ is a permutation matrix and $\boldsymbol{\Sigma} = \mathrm{Diag}(r_1, \ldots r_{d_C}) \in \mathbb{R}^{d_C \times d_C}$ is a diagonal scaling matrix.

Finally, since $c_i \overset{(d)}{\neq\!=} kc_j, \forall i \neq j, \forall k \in \mathbb{R}$, $\boldsymbol{M}$ has to to be identity matrix. To see the reason, for the sake of contradiction, suppose that either (i) there exist $i, j \in [d_C] \times [d_C]$ and $k \in \mathbb{R}$, with $i \neq j$ such that $[\boldsymbol{M}\boldsymbol{c}]_i = kc_j$, or (ii) $\exists i \in [d_C]$ such that $[\boldsymbol{M}\boldsymbol{c}]_i = kc_i$ for some $k \in \mathbb{R}, k \neq 1$.

For case (i), since $\boldsymbol{M}\boldsymbol{c} \overset{(d)}{=\!=} \boldsymbol{c}$, $[\boldsymbol{M}\boldsymbol{c}]_i \overset{(d)}{=\!=} c_i, \forall i$. Hence,

$$[\boldsymbol{M}\boldsymbol{c}]_i = kc_j$$

$$\implies [\boldsymbol{M}\boldsymbol{c}]_i \overset{(d)}{=\!=} kc_j$$

$$\implies c_i \overset{(d)}{=\!=} kc_j,$$

which is a contradiction to the assumption $c_i \overset{(d)}{\neq} kc_j$.

For case (ii), $[\boldsymbol{M}\boldsymbol{c}]_i = kc_j$ implies that $c_i \overset{(d)}{=\!=} kc_j$. First, $k \neq \pm 1$, cannot hold because it will mean that $\mathrm{var}(c_i) = k^2 \mathrm{var}(c_i)$ which cannot hold for $k \neq \pm 1$ since $\mathrm{var}(c_i) > 0$. Hence, the only possible option is $k = -1$, which is already ruled out by the assumption that $c_i \overset{(d)}{\neq} -c_i$. Hence $\boldsymbol{M}$ is an identity matrix. This concludes the proof.

## B.3 Considering Assumption (b)

Let

$$\boldsymbol{e}_i = [0, 0, \, \ldots, \underset{i\text{th location}}{1}, 0, 0] \in \mathbb{R}^{d_C}$$

denote the standard basis vector in $\mathbb{R}^{d_C}$. Let vertex of hyper-rectangle $\boldsymbol{v}_i = a_i \boldsymbol{e}_i = \boldsymbol{\Lambda}\boldsymbol{e}_i$, where

$$\boldsymbol{\Lambda} = \mathrm{Diag}([a_1, \ldots, a_{d_C}]^T),$$

where $\mathrm{Diag}(\cdot)$ represents the diagonal matrix formed by the given vector.

If $\boldsymbol{M}\boldsymbol{c} \overset{(d)}{=\!=} \boldsymbol{c}$, then the supports of $\boldsymbol{M}\boldsymbol{c}$ and $\boldsymbol{c}$ should match, i.e.,

$$\boldsymbol{M}(\mathcal{C}) = \mathcal{C},$$

where $\boldsymbol{M}(\mathcal{C}) = \{\boldsymbol{M}\boldsymbol{c} \mid \boldsymbol{c} \in \mathcal{C}\}$.

Note that $\forall \boldsymbol{c} \in \mathcal{C}$, the set of points $\boldsymbol{v}_i$ satisfy the following property

$$\boldsymbol{c} = \sum_{i=1}^{d_C} \alpha_i \boldsymbol{v}_i, \qquad \text{for some } -1 \leq \alpha_i \leq 1 \tag{19}$$

$$\implies \boldsymbol{M}\boldsymbol{c} = \sum_{i=1}^{d_C} \alpha_i (\boldsymbol{M}\boldsymbol{v}_i).$$

Since the support of $\boldsymbol{Mc}$ is $\mathcal{C}$, this implies that $\forall \boldsymbol{c} \in \mathcal{C}$,

$$\boldsymbol{c} = \sum_{i=1}^{d_{\mathrm{C}}} \alpha_i (\boldsymbol{Mv}_i), \qquad \text{for some } -1 \le \alpha_i \le 1$$

The last equation implies that the set of points $\boldsymbol{Mv}_i, \forall i \in [d_{\mathrm{C}}]$ also satisfy property (19). Hence, for each $i \in [d_{\mathrm{C}}]$, $\boldsymbol{Mv}_i = \pm \boldsymbol{v}_j$ for some unique $j \in [d_{\mathrm{C}}]$. Note that $j$ should be unique for each $i$ because $\boldsymbol{M}$ is invertible, hence $\boldsymbol{M}$ cannot map two orthogonal vectors $\boldsymbol{v}_i$ and $\boldsymbol{v}_k$, $i \ne k$, to the same vector $\pm \boldsymbol{v}_j$ with same or different signs.

Let $\boldsymbol{V} = [\boldsymbol{v}_1, \ldots, \boldsymbol{v}_{d_{\mathrm{C}}}]^T$. Then one can write

$$\boldsymbol{MV} = \boldsymbol{V\Sigma\Pi},$$

where $\boldsymbol{\Sigma}$ is some diagonal matrix with diagonal entries from $\{+1, -1\}$ and $\boldsymbol{\Pi}$ is a permutation matrix. Then the above implies

$$\boldsymbol{M\Lambda I} = \boldsymbol{\Lambda I \Sigma \Pi}$$
$$\implies \boldsymbol{M} = \boldsymbol{\Lambda \Sigma \Pi \Lambda}^{-1}.$$

Hence $\boldsymbol{M}$ is a permutation and scaling matrix.

Finally, by the same argument presented in last paragraph of Sec. B.2 (i.e., proof with Assumption (a)), we conclude that $\boldsymbol{M}$ is an identity matrix.

## C   Proof of Theorem 2

We restate the theorem here:

**Theorem** 2 Consider the mixture model in (8). Assume that $\mathrm{rank}(\boldsymbol{A}) = d_{\mathrm{C}} + d_{\mathrm{P}}$ and $\mathrm{rank}(\mathbb{E}[\boldsymbol{cc}^\top]) = d_{\mathrm{C}}$, and that Assumption 2 holds. Denote $\widehat{\boldsymbol{Q}}$ as any solution of (6) by constraining $\boldsymbol{Q} = \boldsymbol{Q}^{(1)} = \boldsymbol{Q}^{(2)}$. Then, we have $\widehat{\boldsymbol{Q}}\boldsymbol{x}^{(q)} = \boldsymbol{\Theta c}$.

One can follow the same argument as in the step 1 of proof in B.

Let us define

$$\boldsymbol{H} = \boldsymbol{QA} \in \mathbb{R}^{d_{\mathrm{C}} \times (d_{\mathrm{C}} + d_{\mathrm{P}})}.$$

We want to show that

$$\mathrm{Null}(\boldsymbol{H}) = \boldsymbol{0} \times \mathbb{R}^{d_{\mathrm{P}}}, \tag{20}$$

since this will imply that that $\boldsymbol{H}$ does not depend upon the style component. Combined with the fact that $\mathrm{rank}(\boldsymbol{H}) = d_{\mathrm{C}}$, this will imply that $\boldsymbol{H}$ is an invertible function of the content component. To that end, consider the following line of arguments.

Since the objective in (6) matches the distribution for latent random variables $\widehat{\boldsymbol{c}}^{(1)} = \boldsymbol{Qx}^{(1)}$ and $\widehat{\boldsymbol{c}}^{(2)} = \boldsymbol{Qx}^{(2)}$, the following holds for any $\mathcal{R}_c \subseteq \mathbb{R}^{d_C}$,, $\exists k \in \mathbb{R}$

$$\mathbb{P}_{\widehat{\boldsymbol{c}}^{(1)}}[k\mathcal{R}_c] = \mathbb{P}_{\widehat{\boldsymbol{c}}^{(2)}}[k\mathcal{R}_c],$$

$$\overset{(a)}{\Longleftrightarrow} \mathbb{P}_{\boldsymbol{z}^{(1)}}[\boldsymbol{H}_{\mathrm{PreImg}}(k\mathcal{R}_c)] = \mathbb{P}_{\boldsymbol{z}^{(2)}}[\boldsymbol{H}_{\mathrm{PreImg}}(k\mathcal{R}_c)] \tag{21}$$

$$\overset{(b)}{\Longleftrightarrow} \mathbb{P}_{\boldsymbol{z}^{(1)}}[k\boldsymbol{H}_{\mathrm{PreImg}}(\mathcal{R}_c)] = \mathbb{P}_{\boldsymbol{z}^{(2)}}[k\boldsymbol{H}_{\mathrm{PreImg}}(\mathcal{R}_c)], \tag{22}$$

where, $\boldsymbol{H}_{\mathrm{PreImg}}(\mathcal{R}_c) := \{\boldsymbol{z} \mid \boldsymbol{Hz} \in \mathcal{R}_c\}$ is the pre-image of $\boldsymbol{H}$. $(a)$ follows because $\mathbb{P}_{\widehat{\boldsymbol{c}}^{(q)}}[k\mathcal{R}_c] = \mathbb{P}_{\boldsymbol{Hz}^{(q)}}[k\mathcal{R}_c] = \mathbb{P}_{\boldsymbol{z}^{(q)}}[\boldsymbol{H}_{\mathrm{PreImg}}(k\mathcal{R}_c)]$ [68, Section 2.2]. $(b)$ follows because $\boldsymbol{H}$ is a linear operation.

Although (21) holds for any $\mathcal{R}_c$, we will see that it is sufficient to consider a special $\mathcal{R}_c$ to prove (20). To that end, take $\mathcal{R}_c = \mathrm{conv}\{\boldsymbol{0}, \boldsymbol{a}_1, \ldots, \boldsymbol{a}_{d_{\mathrm{C}}}\}$, where $\boldsymbol{a}_i \in \mathbb{R}^{d_C}$ such that $\mathbb{P}_{\widehat{\boldsymbol{c}}^{(q)}}[\mathcal{R}_c] > 0$. Let us take $\boldsymbol{y}_i \in \mathbb{R}^{d_{\mathrm{C}} + d_{\mathrm{P}}}$, such that $\boldsymbol{Hy}_i = \boldsymbol{a}_i$. For reasons that will be clear later, we hope to show that

$$\boldsymbol{H}_{\mathrm{PreImg}}(\mathcal{R}_c) = \mathrm{conv}\{\boldsymbol{0}, \boldsymbol{y}_1, \ldots, \boldsymbol{y}_{d_{\mathrm{C}}}\} + \mathrm{Null}(\boldsymbol{H}).$$

To that end, observe that for any $\boldsymbol{r} \in \mathcal{R}_c$, we can represent $\boldsymbol{r}$ as,

$$\boldsymbol{r} = \frac{1}{d_{\mathrm{C}} + 1} \sum_{i=1}^{d_{\mathrm{C}}} w_i \boldsymbol{a}_i, \text{ for some } \{w_i\}_{i=1}^{d_{\mathrm{C}}} \text{ s.t. } \sum_{i=1}^{d_{\mathrm{C}}} w_i \leq 1, \ \forall i.$$

For both view $q = 1, 2$, we get,

$$\boldsymbol{r} = \frac{1}{d_{\mathrm{C}} + 1} \sum_{i=1}^{d_{\mathrm{C}}} w_i \boldsymbol{H} \boldsymbol{y}_i$$

$$\implies \boldsymbol{r} = \boldsymbol{H} \left( \frac{1}{d_{\mathrm{C}} + 1} \sum_{i=1}^{d_{\mathrm{C}}} w_i \boldsymbol{y}_i \right)$$

$$\boldsymbol{H}_{\mathrm{PreImg}} \left( \frac{1}{d_{\mathrm{C}} + 1} \sum_{i=1}^{d_{\mathrm{C}}} w_i \boldsymbol{a}_i \right) = \frac{1}{d_{\mathrm{C}} + 1} \sum_{i=1}^{d_{\mathrm{C}}} w_i \boldsymbol{y}_i + \mathrm{Null}(\boldsymbol{H}) \tag{23}$$

We can write,

$$\boldsymbol{H}_{\mathrm{PreImg}}(\mathcal{R}_c) = \mathrm{conv}\{\boldsymbol{0}, \boldsymbol{y}_1, \ldots, \boldsymbol{y}_{d_{\mathrm{C}}}\} + \mathrm{Null}(\boldsymbol{H}) \tag{24}$$

We have that $\mathrm{Null}(\boldsymbol{H}) \subset \mathbb{R}^{d_{\mathrm{C}} + d_{\mathrm{P}}}$ is a linear subspace with $\dim(\mathrm{Null}(\boldsymbol{H})) = d_{\mathrm{P}}$. Let $\mathcal{A} = \boldsymbol{H}_{\mathrm{PreImg}}(\mathcal{R}_c)$. Note that $\mathbb{P}_{\boldsymbol{z}^{(1)}}[k\mathcal{A}] = \mathbb{P}_{\boldsymbol{z}^{(2)}}[k\mathcal{A}], \forall k \in \mathbb{R}$ (from (21), and $\mathbb{P}_{\boldsymbol{z}^{(q)}}[\mathcal{A}] > 0$ (by the construction of $\mathcal{R}_c$). Further, the set $\mathcal{A}$ is of the form

$$\mathrm{conv}\{\boldsymbol{0}, \boldsymbol{y}_1, \ldots, \boldsymbol{y}_{d_{\mathrm{C}}}\} + \mathcal{P},$$

because $\mathrm{Null}(\boldsymbol{H})$ is a subspace of dimension $d_{\mathrm{P}}$, hence it satisfies the definition of $\mathcal{P}$. Hence, Assumption 2 implies that

$$\mathrm{Null}(\boldsymbol{H}) = \boldsymbol{0} \times \mathbb{R}^{d_{\mathrm{P}}}.$$

Denoting the $N$th to $M$th columns of $\boldsymbol{H}$ by $\boldsymbol{H}(N : M)$, the above is equivalent to saying

$$\boldsymbol{H}(d_{\boldsymbol{C}} + 1 : d_{\boldsymbol{C}} + d_{\boldsymbol{P}}) = 0. \tag{25}$$

Denote,

$$\boldsymbol{\Theta} = \boldsymbol{H}(1 : d_{\boldsymbol{C}}) \ \forall \ v = 1, 2.$$

Then, we can write,

$$\boldsymbol{Q}\boldsymbol{x}^{(q)} = \boldsymbol{\Theta}\boldsymbol{c}, \ \forall \ v = 1, 2. \tag{26}$$

This concludes the proof.

## D  Proof of Theorem 3

We restate the theorem here:

**Theorem** 3 Assume that Assumption 1 is satisfied, that $|\mathcal{L}| \geq d_{\mathrm{C}}$ paired samples $(\boldsymbol{x}_\ell^{(1)}, \boldsymbol{x}_\ell^{(2)})$ are available, that $\boldsymbol{A}^{(q)}, \ q = 1, 2$ have full column rank, and that $\mathbb{P}_{\boldsymbol{c}}$ is absolutely continuous. Denote $(\widehat{\boldsymbol{Q}}^{(1)}, \widehat{\boldsymbol{Q}}^{(2)})$ as any optimal solution of (6) under the constraint (9). Then, we have $\widehat{\boldsymbol{Q}}^{(q)} \boldsymbol{x}^{(q)} = \boldsymbol{\Theta}\boldsymbol{c}$.

From our objective in (6), we obtain

$$\boldsymbol{Q}^{(1)} \boldsymbol{x}^{(1)} \overset{\text{(d)}}{=\!=\!=} \boldsymbol{Q}^{(2)} \boldsymbol{x}^{(2)}. \tag{27}$$

Using Assumption 1 and following the proof of step 1 in Theorem B, we can obtain:

$$\boldsymbol{Q}^{(q)} \boldsymbol{x}^{(q)} = \boldsymbol{\Theta}^{(q)} \boldsymbol{c}, \ \forall q = 1, 2,$$

for some invertible matrices $\boldsymbol{\Theta}^{(q)}, \forall q$. Hence,

$$\boldsymbol{\Theta}^{(1)} \boldsymbol{c} \overset{\text{(d)}}{=\!=\!=} \boldsymbol{\Theta}^{(2)} \boldsymbol{c} \tag{28}$$

$$\implies \boldsymbol{c} \overset{\text{(d)}}{=\!=\!=} (\boldsymbol{\Theta}^{(1)})^{-1} \boldsymbol{\Theta}^{(2)} \boldsymbol{c}. \tag{29}$$

Hence we can have linear transformation $\boldsymbol{M} := (\boldsymbol{\Theta}^{(1)})^{-1} \boldsymbol{\Theta}^{(2)}$ which has same probability density as $\mathbb{P}_{\boldsymbol{c}}$. However, the sample matching constraint (9), for $\ell-$th sample implies that

$$\boldsymbol{Q}^{(1)} \boldsymbol{x}_\ell^{(1)} = \boldsymbol{Q}^{(2)} \boldsymbol{x}_\ell^{(2)}$$
$$\implies \boldsymbol{\Theta}^{(1)} \boldsymbol{c}_\ell = \boldsymbol{\Theta}^{(2)} \boldsymbol{c}_\ell$$
$$\implies \boldsymbol{c}_\ell = (\boldsymbol{\Theta}^{(1)})^{-1} \boldsymbol{\Theta}^{(2)} \boldsymbol{c}_\ell$$
$$\implies \boldsymbol{c}_\ell = \boldsymbol{M} \boldsymbol{c}_\ell.$$

Let $\boldsymbol{C} = [\boldsymbol{c}_1 \ldots \boldsymbol{c}_{N_p}]$. Then the above implies:

$$\boldsymbol{C} = \boldsymbol{M} \boldsymbol{C}$$
$$\implies (\boldsymbol{M} - \boldsymbol{I}) \boldsymbol{C} = \boldsymbol{0}.$$

Now we show that $\boldsymbol{C}$ is a full row rank matrix, which implies that $\boldsymbol{M} - \boldsymbol{I} = \boldsymbol{0} \implies \boldsymbol{M} = \boldsymbol{I}$. To that end, note that random variables $\boldsymbol{x}_i^{(1)}$ and $\boldsymbol{x}_i^{(2)}$ being i.i.d implies that $\boldsymbol{c}^{(i)}$ are i.i.d from $\mathbb{P}_{\boldsymbol{c}}$. This implies that for any $1 \le i \le |\mathcal{L}|$,

$$\Pr[\boldsymbol{c}_i \in \text{span}(\{\boldsymbol{c}_{n_1}, \ldots, \boldsymbol{c}_{n_{d_{\mathrm{C}}-1}}\})] = 0. \tag{30}$$

This is because $\text{span}(\{\boldsymbol{c}_{n_1}, \ldots, \boldsymbol{c}_{n_{d_{\mathrm{C}}-1}}\})$ for $n_j \in [|\mathcal{L}|]$, is a lower dimensional subspace in $\mathbb{R}^{d_{\mathrm{C}}}$, which has zero probability under absolutely continuous distribution $\mathbb{P}_{\boldsymbol{c}}$. Hence any $d_{\mathrm{C}}$ out of $|\mathcal{L}|$ column vectors in $\boldsymbol{C}$ are linearly independent with probability 1.

This concludes the proof.

# E    Detailed Identifiability Conditions of Existing Results

## E.1    Identifiability of CCA

> **Theorem 4** (Identifiability of Aligned SCA via CCA [1]). *Under* (1), *assume that every aligned pair* $(\boldsymbol{x}^{(1)}, \boldsymbol{x}^{(2)})$ *share the same* $\boldsymbol{c}$, *and that* $\boldsymbol{A}^{(q)}$ *has full column rank. Also assume that there exists an $N$-sample set* $\{\ell_1, \ldots, \ell_N\}$ *such that* $[\boldsymbol{C}^{\mathsf{T}}, (\boldsymbol{P}^{(1)})^{\mathsf{T}}, (\boldsymbol{P}^{(2)})^{\mathsf{T}}]^{\mathsf{T}} \in \mathbb{R}^{N \times (d_{\mathrm{C}} + d_{\mathrm{P}}^{(1)} + d_{\mathrm{P}}^{(2)})}$ *has full column rank, where* $\boldsymbol{C} = [\boldsymbol{c}_{\ell_1}, \ldots \boldsymbol{c}_{\ell_N}] \in \mathbb{R}^{d_{\mathrm{C}} \times N}$ *and* $\boldsymbol{P}^{(q)} = [\boldsymbol{p}_{\ell_1}^{(q)} \ldots \boldsymbol{p}_{\ell_N}^{(q)}] \in \mathbb{R}^{d_{\mathrm{P}}^{(q)} \times N}$ *for* $q = 1, 2$. *Denote* $(\widehat{\boldsymbol{Q}}^{(1)}, \widehat{\boldsymbol{Q}}^{(2)})$ *as an optimal solution of the CCA formulation. Then, we we have*
>
> $$\widehat{\boldsymbol{Q}}^{(q)} \boldsymbol{x}^{(q)} = \boldsymbol{\Theta} \boldsymbol{c},$$
>
> *where* $\boldsymbol{\Theta}$ *is nonsingular.*

In the above theorem, one can see that $N \ge (d_{\mathrm{C}} + d_{\mathrm{P}}^{(1)} + d_{\mathrm{P}}^{(2)})$ is a *necessary condition* for the identifiability of $\boldsymbol{\Theta} \boldsymbol{c}$. Hence, CCA needs at least $d_{\mathrm{C}} + d_{\mathrm{P}}^{(1)} + d_{\mathrm{P}}^{(2)}$ paired samples for identifiability.

## E.2    Identifiability of Unaligned SCA in [8]

We summarize the result in [8] in the following

> **Theorem 5** (Identifiability of Unaligned SCA via ICA [8]). *Under* (1), *assume that the following are met: (i) The conditions for ICA identifiability [33] is met by each modality, including that the components of* $\boldsymbol{z}^{(q)} = [\boldsymbol{c}^{\mathsf{T}}, (\boldsymbol{p}^{(q)})^{\mathsf{T}}]^{\mathsf{T}}$ *are mutually statistically independent and contain at most one Gaussian variable. In addition, each* $\boldsymbol{z}_i^{(q)}$ *has unit variance; (ii)* $\mathbb{P}_{z_i^{(q)}} \ne \mathbb{P}_{z_j^{(q)}}, \mathbb{P}_{z_i^{(q)}} \ne \mathbb{P}_{-z_j^{(q)}} \forall i, j \in [d_{\mathrm{C}} + d_{\mathrm{P}}^{(q)}], i \ne j$. *Then, assume that* $(i_m, j_m)$ *are obtained by ICA followed by cross domain matching (see the part on Unaligned SCA in Section 2 ) for* $m = 1, \ldots, d_{\mathrm{C}}$.

*Denote $\widehat{c}_m^{(1)} = e_{i_m}^\top \widehat{z}^{(1)}$ and $\widehat{c}_m^{(2)} = e_{j_m}^\top \widehat{z}^{(2)}$. We have the following:*

$$\widehat{c}_m^{(q)} = k c_{\pi(m)}^{(q)}, \; m \in [d_{\mathrm{C}}], \tag{31}$$

*where $k \in \{+1, -1\}$ and $\pi$ is a permutation of $\{1, \dots, d_{\mathrm{C}}\}$.*

## F  Additional Synthetic Data Experiments

**Hyperparameter Settings:**  We use Adam optimizer [70] to solve (7) and learn matrices $Q^{(q)}$, $q = 1, 2$ and the discriminator $f$. We set the initial learning rate of matrix and discriminator to be 0.009 and 0.00008 respectively. We set the $\lambda = 0.1$ in (7) to enforce (6c). For weak supervision experiment in F, we set $\beta = 0.01$ in (9). We generate total of 100,000 samples in each domain. For our experiment we set the batch size to be 1,000 and run (7) for 50 epochs. Our discriminator is a 6-layer multilayer perceptron (MLP) with hidden units { 1024, 521, 512, 256, 128, 64 } in each layer. All the layers use leaky ReLU activation functions [71] with a slope of 0.2 except for the last layer which has sigmoid activations. We include a label smoothing coefficient of 0.2 in the discriminator predictions as suggested in [40].

**Additional Details for Validation of Theorem 1 in Sec. 3:**  Here we explain the data generation details of the result shown in Fig. 3. For the result in top row, we sample $c_1$ from a Gaussian mixture with three Gaussian components. Each component follows a normal distribution $\mathcal{N}(\mu,\ 2)$ where $\mu \sim \mathcal{N}(0,\ 10)$. The second component, i.e., $c_2$, is independently sampled from the gamma distribution $\texttt{Gamma}(1,\ 3)$. The private components are sampled from $p^{(1)} \sim \texttt{Laplace}(1.0,\ 6.5)$ and $p^{(2)} \sim \texttt{Uniform}[-10,\ 10]$, both only having one dimension. In the bottom row, we sample $c \in \mathbb{R}^2 \sim \texttt{VonMises}(2.5,\ 2.0)$ distribution. The private components satisfy $p^{(1)} \sim \texttt{Laplace}(1.0,\ 6.5)$ and $p^{(2)} \sim \texttt{Gamma}(0.5,\ 3.0)$. Each element of mixing matrices are sampled from $A_{ij}^{(q)} \sim \mathcal{N}(0,1)$, $q = 1, 2$. The readers are referred to Table 4 for the definition of notations used for distributions.

**Validation of Theorem 1 under different sample sizes and imbalanced data:**  Here we observe the shared component identification performance of the proposed method numerically. We conducted two experiments in different settings. First, we vary the sample sizes in both modalities, but the two modalities have the same sample size. Second, we only vary the sample size of modality 2 while keeping the sample size of modality 1 fixed. This way, we create the data imbalance between modalities. Note that the shared components are identified if the following two conditions are met:

1. $Q^{(q)} A^{(q)} = [\Theta, 0]$, i.e., $\widehat{\Theta}^{(1)} = \widehat{\Theta}^{(2)} = \Theta$ and
2. $\|Q^{(q)} A^{(q)} (d_{\mathrm{C}} : d_{\mathrm{C}} + d_{\mathrm{P}}^{(q)})\|_{\mathrm{F}} = 0$.

Therefore, we use the above as our performance evaluation metrics. For the following experiments (Table 5 and 6), we generate the data for the two modalities by sampling a two-dimensional content $c \sim \texttt{VonMises}(2.5,\ 2.0)$ and private components from $p^{(1)} \sim \texttt{Laplace}(1.0,\ 6.5)$ and $p^{(2)} \sim \texttt{Gamma}(0.5,\ 3.0)$. The elements of the mixing matrices are sampled as $A_{ij}^{(q)} \sim \mathcal{N}(0,1)$, $q = 1, 2$. We report the mean and standard deviation of $\|\widehat{\Theta}^{(1)}(1 : d_{\mathrm{C}}) - \widehat{\Theta}^{(2)}(1 : d_{\mathrm{C}})\|_{\mathrm{F}}$ and $1/2 \sum_{q=1}^2 \|\widehat{\Theta}^{(q)}(d_{\mathrm{C}} : d_{\mathrm{C}} + d_{\mathrm{P}}^{(q)})\|_{\mathrm{F}}$ obtained from 5 different runs.

Table 5 shows the performance of SCA and CCA under different sample sizes (i.e., $N$). One can see that the proposed method (SCA) clearly identifies the shared components even when only 100 samples are available. The performance starts to deteriorate when $N \leq 50$, probably because the min-max optimization problem is difficult to solve with very few samples. CCA does not really work under this setting as it needs aligned cross-domain samples.

Table 6 shows the performance of SCA in the cases where two modalities have unbalanced data sizes. The number of samples in the first modality is fixed to 100,000 while the second modality's data size varies from 10,000 to 10 samples. The data generation process remains the same as in the previous experiment. One can see that even under obvious cross-domain data size imbalance (e.g., 100,000 to 1,000), the proposed method performs reasonably well in terms of shared component identification.

Table 5: Shared component identification performance over different $N$.

| $N$ | $\|\widehat{\Theta}^{(1)}(1:d_{\mathrm{C}}) - \widehat{\Theta}^{(2)}(1:d_{\mathrm{C}})\|_{\mathrm{F}}$ | | $1/2\sum_{q=1}^2 \|\widehat{\Theta}^{(q)}(d_{\mathrm{C}}:d_{\mathrm{C}}+d_{\mathrm{P}}^{(q)})\|_{\mathrm{F}}$ | |
| --- | --- | --- | --- | --- |
| | SCA | CCA | SCA | CCA |
| 100,000 | $0.015 \pm 0.020$ | $1.623 \pm 0.273$ | $0.031 \pm 0.009$ | $0.232 \pm 0.033$ |
| 10,000 | $0.021 \pm 0.006$ | $1.667 \pm 0.240$ | $0.030 \pm 0.002$ | $0.267 \pm 0.031$ |
| 1,000 | $0.018 \pm 0.011$ | $1.572 \pm 0.474$ | $0.042 \pm 0.059$ | $0.280 \pm 0.070$ |
| 100 | $0.053 \pm 0.014$ | $2.224 \pm 0.525$ | $0.083 \pm 0.096$ | $0.364 \pm 0.153$ |
| 50 | $0.132 \pm 0.118$ | $1.469 \pm 0.299$ | $0.142 \pm 0.132$ | $1.470 \pm 1.520$ |
| 20 | $1.373 \pm 0.626$ | $2.084 \pm 0.661$ | $0.490 \pm 0.321$ | $0.546 \pm 0.269$ |

Table 6: Shared component identification performance under imbalanced multi-modal data sizes.

| # samples in modality 2 | $\|\widehat{\Theta}^{(1)}(1:d_{\mathrm{C}}) - \widehat{\Theta}^{(2)}(1:d_{\mathrm{C}})\|_{\mathrm{F}}$ | $\frac{1}{2}\sum_{q=1}^2 \|\widehat{\Theta}^{(q)}(d_{\mathrm{C}}:d_{\mathrm{C}}+d_{\mathrm{P}}^{(q)})\|_{\mathrm{F}}$ |
| --- | --- | --- |
| 10,000 | $0.020 \pm 0.018$ | $0.018 \pm 0.005$ |
| 1,000 | $0.065 \pm 0.029$ | $0.026 \pm 0.015$ |
| 100 | $0.145 \pm 0.051$ | $0.081 \pm 0.049$ |
| 10 | $1.290 \pm 0.239$ | $0.293 \pm 0.064$ |

**Validation of Theorem 3.** Fig. 5 presents numerical validation for Theorem 3.

*Data Generation:* We set $d_{\mathrm{C}} = 3$ and $d_{\mathrm{P}}^{(q)} = 1$ for $q = 1, 2$. We sample each component of shared component $c_i \sim \mathtt{Laplace}(0.0,\ 6.5)$ $i = 1, 2, 3$, $p^{(1)} \in \mathbb{R}^1 \sim \mathtt{Uniform}[-10,\ 10]$ and $p^{(2)} \sim \mathtt{Gamma}(0.5,\ 3.0)$. Although $c$ satisfies component-wise independence assumption, it does not satisfy the condition that $c_i \overset{(\mathrm{d})}{\neq} kc_j, \forall i \neq j$ because $c_i \overset{(\mathrm{d})}{=\!=} c_j, \forall i, j \in [3]$. Therefore, Theorem 1 does not cover this case. Nonetheless, this case falls under the jurisdiction of Theorem 3.

*Result:* Fig. 5 corroborates with our Theorem 3. That is, one needs at least $|\mathcal{L}| \geq d_{\mathrm{C}} = 3$ pairs of "anchors" (i.e., aligned cross domain pairs) to ensure identifiability of $\widehat{c}^{(q)} = \Theta c$ for $q = 1, 2$.

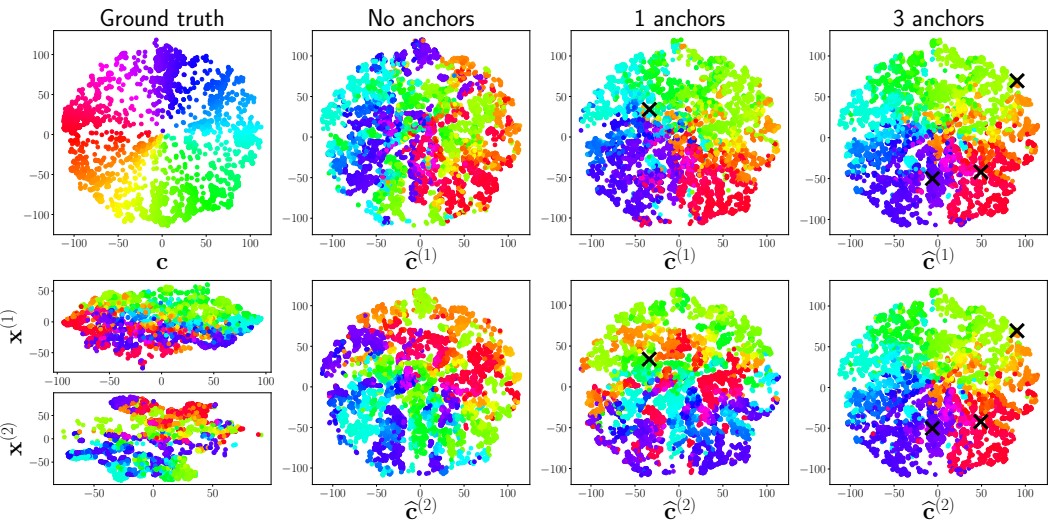

Figure 5: Validation of Theorem 3 $d_{\mathrm{C}} = 3$ and $d_{\mathrm{P}}^{(1)} = 1$.

# G   Real Data Experiment Settings and Additional Results

## G.1   Domain Adaptation

**Hyperparameter Settings:** The domain adaptation task follows the hyperparameter settings described in Table. 7.

Table 7: Hyperparameter settings for domain adaptation.

| Parameter | Value |
|---|---|
| Optimizer | Adam |
| Learning rate of $\boldsymbol{Q}$ | 0.0002 |
| Learning rate of $f$ | 0.00002 |
| Learning rate of classifier | 0.02 |
| Learning rate decay of classifier | 0.75 |
| $\lambda$ (see Eq. (7)) | 1.0 |
| $\gamma$ (see Eq. (32)) | 0.1 |
| Batch size | 64 |
| Number of epochs | 20 |
| Discriminator; $f$ architecture | 6 layers, hidden units {1024, 521, 512, 256, 128, 64} |
| Activation functions of $f$ | Leaky ReLU (slope 0.2), Sigmoid (final layer) |
| Label smoothing coefficient in $f$ | 0.2 |

Table 8: Classification accuracy on the target domain of *office-31* dataset using CLIP embeddings.

| source $\rightarrow$ target | CLIP | DANN | MDD | MCC | SDAT | SDAT+MCC | ELS | ELS+MCC | Proposed | Proposed+MCC |
|---|---|---|---|---|---|---|---|---|---|---|
| **A $\rightarrow$ W** | 93.4 | 93.7 | 94.1 | 95.9 | 95.0 | 98.1 | 96.8 | **98.7** | 95.3 | 98.3 |
| **D $\rightarrow$ W** | 99.1 | **100.0** | 99.3 | **100.0** | **100.0** | **100.0** | **100.0** | **100.0** | 99.7 | **100.0** |
| **W $\rightarrow$ D** | **100.0** | 99.5 | 99.5 | 98.4 | 99.5 | 99.5 | 99.5 | **100.0** | **100.0** | 99.9 |
| **A $\rightarrow$ D** | 91.9 | 92.1 | 94.2 | 97.7 | 95.7 | 97.7 | 95.0 | 97.7 | 93.7 | **99.1** |
| **D $\rightarrow$ A** | 81.4 | 81.9 | 79.2 | 85.7 | 81.2 | 84.6 | 81.3 | 83.0 | 83.9 | **85.9** |
| **W $\rightarrow$ A** | 81.7 | 83.0 | 82.2 | 84.7 | 84.7 | 86.7 | 82.6 | 86.3 | 85.8 | **87.1** |
| **Average** | 91.2 | 91.6 | 91.4 | 93.7 | 92.6 | 94.4 | 92.5 | 94.2 | 93.0 | **95.0** |

**Baselines and Training Setup**: The baselines are representative DA methods, namely, DANN [25], MDD [60], MCC [61], SDAT [62], and ELS [63]. We use the implementations of DANN, MDD, and MCC from the `https://github.com/thuml/Transfer-Learning-Library`, while SDAT and ELS are taken from `https://github.com/yfzhang114/Environment-Label-Smoothing`. In all the baselines, the classifier is jointly optimized with the feature extractor $\boldsymbol{Q}$ which arguably regularizes towards more classification-friendly geometry of the shared features; see [72, 73]. Following their training strategy, we also append a cross-entropy (CE) based classifier training module to our loss in (7) (which learns our feature extractor $\boldsymbol{Q}$). The CE part uses $\boldsymbol{Q}\boldsymbol{x}^{(1)}$ and the labels of the sources as inputs to learn the classifier, i.e.,

$$\mathcal{L}_{\mathrm{CE}} = -\gamma \sum_{\ell=1}^{N} \sum_{k=1}^{K} \mathbb{I}[y_\ell = k] \log \boldsymbol{r}_{\boldsymbol{\theta}}([\boldsymbol{Q}\boldsymbol{x}_\ell^{(1)}]_k), \tag{32}$$

where $\boldsymbol{r}_{\boldsymbol{\theta}}(\cdot) : \mathbb{R}^{d_C} \rightarrow \mathbb{R}^K$ is the classifier that aims to map the learned feature vector $\boldsymbol{Q}\boldsymbol{x}_\ell^{(1)}$ to a $K$-dimensional probability mass function (i.e., the distribution of the ground-truth label over $K$ classes), $y_\ell \in [K]$ represents the label of the $\ell$th sample in source domain, and the indicator function $\mathbb{I}[y_\ell = k] = 1$ only when the event $y_\ell = k$ happens (other wise $\mathbb{I}[y_\ell = k] = 0$. The $\gamma \geq 0$ is the tunable parameter. The joint loss is still differentiable, and thus we still use the Adam optimizer to jointly optimize $\boldsymbol{Q}$ and $\boldsymbol{\theta}$.

**Additional domain adaptation experiment using CLIP features:** In this experiment, we use CLIP as an image encoder as it learns informative and transferable features from very large datasets [35]. Table 8 and Table 9 show the results on *Office-31* and *Office-Home* datasets, respectively, using CLIP embeddings. Compared to the results on ResNet50 embeddings in Table 1 and Table 2, one can observe that all the methods, including proposed method, gains an advantage. This is likely because CLIP was trained on a large and diverse dataset [35], which may have include similar content to the *Office-31* and *Office-Home* datasets.

The results show that, as a foundation model, CLIP can already unify the embeddings of the source and target domains to a reasonable extent. In addition, our model and algorithm when combined with regularization techniques like MCC, can still further enhance performance, even with simple post-processing of CLIP embeddings.

Table 9: Classification accuracy on the target domain of *office-Home* dataset using CLIP embeddings.

| source → target | CLIP | DANN | MDD | MCC | SDAT | SDAT+MCC | ELS | ELS+MCC | Proposed | Proposed+MCC |
|---|---|---|---|---|---|---|---|---|---|---|
| **Ar → Cl** | 78.0 | 80.4 | 80.2 | 80.9 | 79.6 | 80.7 | 80.0 | 81.3 | 82.0 | **83.2** |
| **Ar → Pr** | 88.7 | 91.7 | 88.9 | 93.3 | 89.4 | 94.3 | 91.2 | 93.9 | 91.4 | **95.2** |
| **Ar → Rw** | 90.6 | 90.2 | 91.0 | 92.8 | 90.1 | 92.1 | 89.4 | 92.1 | 91.9 | **93.8** |
| **Cl → Ar** | 85.2 | 83.2 | 85.1 | 87.4 | 83.1 | 86.1 | 84.4 | 87.2 | 85.4 | **87.7** |
| **Cl → Pr** | 89.0 | 89.7 | 90.1 | 93.4 | 90.2 | 93.5 | 89.7 | 93.5 | 91.1 | **94.9** |
| **Cl → Rw** | 89.8 | 88.1 | 89.4 | 89.3 | 87.9 | 90.5 | 88.3 | 90.6 | 90.4 | **92.0** |
| **Pr → Ar** | 78.2 | 80.4 | 81.8 | 83.7 | 81.0 | 85.0 | 81.8 | 86.1 | 83.0 | **86.6** |
| **Pr → Cl** | 72.7 | 75.8 | 75.8 | 78.4 | 75.4 | 78.5 | 75.7 | 78.3 | 77.7 | **81.3** |
| **Pr → Rw** | 89.0 | 90.4 | 90.3 | 92.6 | 90.8 | 92.1 | 90.0 | 92.3 | 91.0 | **93.6** |
| **Rw → Ar** | 86.6 | 84.9 | 85.9 | 85.3 | 85.3 | 86.3 | 85.4 | 87.0 | 87.7 | **88.2** |
| **Rw → Cl** | 78.1 | 79.4 | 79.8 | 79.0 | 78.6 | 79.8 | 79.1 | 79.8 | 81.3 | **81.8** |
| **Rw → Pr** | 94.3 | 94.6 | 93.9 | 95.9 | 94.8 | 95.4 | 94.0 | 95.2 | 94.7 | **96.0** |
| **Average** | 85.0 | 85.7 | 86.0 | 87.6 | 85.5 | 87.8 | 85.7 | 88.1 | 87.3 | **89.5** |

**Visualization Result:** Fig. 6 shows the 2-dimensional visualization of the CLIP-learned features ($d = 256$) from two domains, namely, DSLR and Amazon images (*Office-31*), using t-SNE. One can see that CLIP could roughly group the same classes from the two domains together. But the proposed method can further pull the circles and the triangle markers together—meaning that the $Q$ really learns shared representations of the same data in the DSLR and Amazon domains.

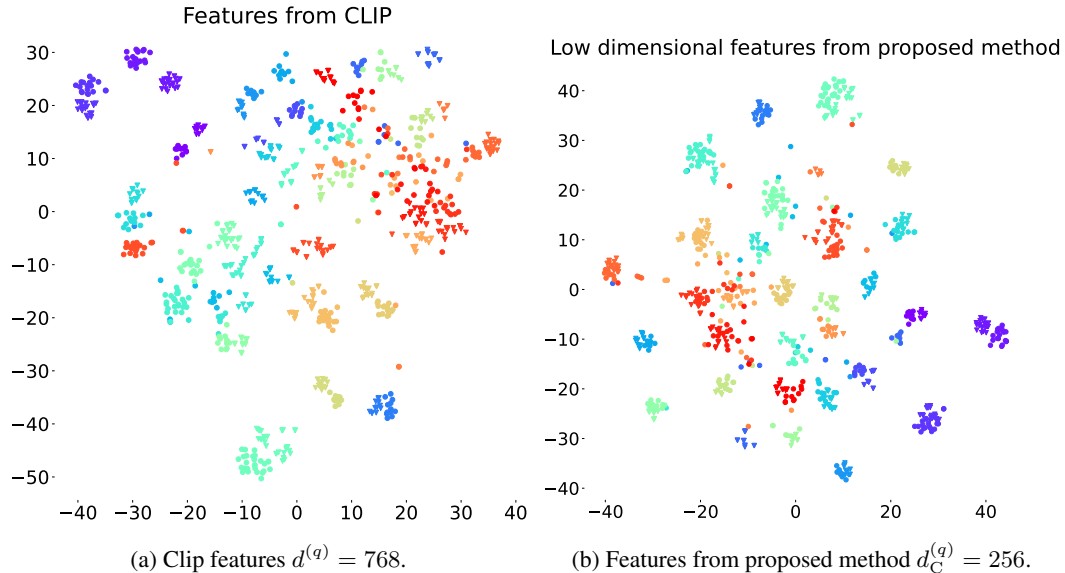

Features from CLIP

Low dimensional features from proposed method

(a) Clip features $d^{(q)} = 768$.

(b) Features from proposed method $d_C^{(q)} = 256$.

Figure 6: *Office-31* dataset: DSLR images features represented as circle markers, Amazon images features represented as triangle markers. Different color represent different classes.

## G.2 Single-cell Sequence Analysis

**Hyperparameter Settings:** The hyperparameter settings for single-cell sequence analysis is presented in Table. 10.

**Baseline:** For more details on baseline refer to the implementation in `https://github.com/uhlerlab/cross-modal-autoencoders`.

## G.3 Multi-lingual Information Retrieval

**Hyperparameter Settings:** The hyperparameter settings for multi-lingual information retrieval is described in the Table. 11.

Table 10: Hyperparameter settings for single-cell sequence analysis.

| Parameter | Value |
|---|---|
| Optimizer | Adam |
| Learning rate of $\boldsymbol{Q}^{(q)}$ | 0.001 |
| Learning rate of $f$ | 0.0001 |
| $\lambda$ (see Eq. (7)) | 1.0 |
| $\beta$ (see Eq. (9)) | 10.0 |
| Batch size | 32 |
| Number of epochs | 75 |
| Discriminator; $f$ architecture | 6 layers, hidden units {1024, 521, 512, 256, 128, 64} |
| Activation functions of $f$ | Leaky ReLU (slope 0.2), Sigmoid (final layer) |
| Label smoothing coefficient in $f$ | 0.2 |

Table 11: Hyperparameter settings for multi-lingual information retrieval.

| Parameter | Value |
|---|---|
| Optimizer | Adam |
| Learning rate of $\boldsymbol{Q}$ | 0.0001 |
| Learning rate of $f$ | 0.00001 |
| $\lambda$ (see Eq. (7)) | 1.0 |
| Batch size | 32 |
| Number of epochs | 5 |
| Discriminator; $f$ (similar as in [21]) | 2 layers, 2048 hidden units each |
| Activation functions of $f$ | Leaky ReLU (slope 0.2), Sigmoid (final layer) |
| Dropout rate (Input) in $f$ | 0.1 |
| Label smoothing coefficient in $f$ | 0.2 |

**Additional Results:** Table 12 reports the P@5 and P@10 scores over the test data, calculated for different source and target language pairs. It can be observed that the proposed method achieves higher precision than Adv in most of the translation tasks (e.g., by at least $1\%$ in the **en→es** and **es→en** tasks) when considering both P@5 and P@10 scores.

### G.4 Computation resources

All the experiments were run on Nvidia H100 GPU. The approximate runtime for a single run of the algorithm is 20 minutes for multi-lingual information retrieval, 15 minutes for domain adaptation, and 3 minutes for single-cell sequence analysis.

**Complexity Analysis:**

Since the proposed objective is tackled using stochastic gradient (SG)-based first-order iterative method, the computational complexity of the proposed algorithm depends upon the per-iteration complexity.

For each sample, the per-iteration complexity is composed of a forward pass and a backward pass.

Note that the problem size depends upon $d^{(q)}$ (the data dimension), $d_C$, and the batch size $B$. We assume that the network architecture of $\boldsymbol{f}$ (the number of layers and hidden units in each layer) is fixed, represented by $\boldsymbol{f} = \boldsymbol{\sigma} \circ \boldsymbol{F}_L \circ \cdots \circ \boldsymbol{\sigma} \circ \boldsymbol{F}_1$, where $\boldsymbol{F}_\ell$ and $\boldsymbol{\sigma}$ are the linear layer (matrix) and activation function corresponding to the $\ell$th layer. Only the input dimension $d_C$ of first matrix $\boldsymbol{F}_1$, varies with the problem size.

The forward pass involves computing $\widehat{\boldsymbol{c}}^{(q)} = \boldsymbol{Q}^{(q)}\boldsymbol{x}^{(q)}$ and $\boldsymbol{f}^{(q)}(\widehat{\boldsymbol{c}}^{(q)})$, both of which scale linearly with $d_C, d^{(q)}$ and the batch size $B$. Hence, the forward pass time complexity is $O(Bd_C(d^{(1)} + d^{(2)}))$.

Similarly, the backward pass requires computing of $\frac{\partial L}{\partial \widehat{\boldsymbol{c}}_i^{(q)}}, \forall i \in [d_C]$ and $\frac{\partial \widehat{\boldsymbol{c}}_i^{(q)}}{\partial \boldsymbol{Q}^{(q)}_{jk}}, \forall i \in [d_C], j, k \in [d_C \times d^{(q)}]$, where $L$ is the loss function. The first gradient computation is linear in $Bd_C$, while the second gradient computation has a complexity of $O(Bd_C(d^{(1)} + d^{(2)}))$. Hence the computational complexity of our method is $O(Bd_C(d^{(1)} + d^{(2)}))$.

Table 12: Average precision P@k of cross-language information retrieval

| P@k | source → target | Adv - NN | proposed - NN | Adv - CSLS | proposed - CSLS |
|---|---|---|---|---|---|
| P@5 | **en→es** | 77.9 | **78.8** | 83.6 | **85.2** |
| | **es→en** | 73.2 | **79.0** | 83.2 | **86.0** |
| | **en→it** | 68.0 | **70.8** | 76.8 | **82.4** |
| | **it→en** | **79.0** | 77.6 | **71.2** | 66.9 |
| | **en→fr** | **79.2** | 75.2 | **85.7** | 85.2 |
| | **fr→en** | 69.8 | **73.5** | 77.2 | **83.5** |
| P@10 | **en→es** | 82.4 | **82.6** | 87.0 | **87.8** |
| | **es→en** | 79.0 | **82.3** | 87.2 | **88.8** |
| | **en→it** | 74.1 | **75.6** | 81.7 | **85.6** |
| | **it→en** | 71.5 | **75.8** | 82.0 | **82.9** |
| | **en→fr** | **83.4** | 79.1 | 88.0 | **88.4** |
| | **fr→en** | 73.8 | **77.4** | 80.6 | **86.6** |

The memory complexity involves storing the network parameters and the aforementioned gradients. Hence, only the size of $\boldsymbol{Q}^{(q)}$, $\boldsymbol{F}_1$, and $\boldsymbol{c}^{(q)}$ changes with the problem dimension. The size of $\boldsymbol{Q}^{(q)}$, $\boldsymbol{F}_1$, and $\boldsymbol{c}^{(q)}$ are $d_{\mathrm{C}}d^{(q)}$, $O(d_{\mathrm{C}})$ and $d_{\mathrm{C}}$, respectively. Therefore, the space complexity is $O(Bd_{\mathrm{C}}(d^{(1)} + d^{(2)}))$.

In summary, both the memory and computational complexities of the proposed method scales linearly with $d_{\mathrm{C}}$.

## H  Extension: Private Component Identification

Theorems 1-3 are concerned with learning the shared component $\boldsymbol{c}$. The goal, there, was to ensure that $\boldsymbol{Q}_{\mathrm{C}}^{(q)}\boldsymbol{x}^{(q)}\boldsymbol{\Theta}\boldsymbol{c}, \forall q$. In some cases, the private components $\boldsymbol{p}^{(q)}$ is also of interest [6, 31, 74]. To learn $\boldsymbol{p}^{(q)}$, we propose to solve the following learning criterion:

$$\text{find} \quad \boldsymbol{Q}_{\mathrm{C}}^{(q)} \in \mathbb{R}^{d_{\mathrm{C}} \times d^{(q)}}, \boldsymbol{Q}_{\mathrm{P}}^{(q)} \in \mathbb{R}^{d_{\mathrm{P}}^{(q)} \times d^{(q)}} \quad q = 1, 2, \tag{33a}$$

$$\text{subject to} \quad \boldsymbol{Q}_{\mathrm{C}}^{(1)}\boldsymbol{x}^{(1)} \overset{\text{(d)}}{=\!=} \boldsymbol{Q}_{\mathrm{C}}^{(2)}\boldsymbol{x}^{(2)}, \tag{33b}$$

$$\boldsymbol{Q}_{\mathrm{C}}^{(q)}\boldsymbol{x}^{(q)} \perp\!\!\!\perp \boldsymbol{Q}_{\mathrm{P}}^{(q)}\boldsymbol{x}^{(q)} \quad q = 1, 2, \tag{33c}$$

$$\boldsymbol{Q}_{\mathrm{C}}^{(q)}\mathbb{E}\left[\boldsymbol{x}^{(q)}(\boldsymbol{x}^{(q)})^{\top}\right](\boldsymbol{Q}_{\mathrm{C}}^{(q)})^{\top} = \boldsymbol{I} \quad q = 1, 2, \tag{33d}$$

$$\boldsymbol{Q}_{\mathrm{P}}^{(q)}\mathbb{E}\left[\boldsymbol{x}^{(q)}(\boldsymbol{x}^{(q)})^{\top}\right](\boldsymbol{Q}_{\mathrm{P}}^{(q)})^{\top} = \boldsymbol{I} \quad q = 1, 2, \tag{33e}$$

where $\boldsymbol{u} \perp\!\!\!\perp \boldsymbol{v}$ means that the random vectors $\boldsymbol{u}$ and $\boldsymbol{v}$ are independent with each other.

For implementation we use following criterion,

$$\min_{\boldsymbol{Q}_{\mathrm{C}}^{(1)}, \boldsymbol{Q}_{\mathrm{C}}^{(2)}\boldsymbol{Q}_{\mathrm{P}}^{(1)}, \boldsymbol{Q}_{\mathrm{P}}^{(1)}} \max_{f} \mathbb{E}_{\boldsymbol{x}^{(1)}} \log\left(f(\boldsymbol{Q}_{\mathrm{C}}^{(1)}\boldsymbol{x}^{(1)})\right) + \mathbb{E}_{\boldsymbol{x}^{(2)}} \log\left(1 - f(\boldsymbol{Q}_{\mathrm{C}}^{(2)}\boldsymbol{x}^{(2)})\right)$$

$$+ \lambda \sum_{q=1}^{2} \mathcal{R}\left(\boldsymbol{Q}_{\mathrm{C}}^{((q))}\right) + \omega \sum_{q=1}^{2} \mathcal{R}\left(\boldsymbol{Q}_{\mathrm{P}}^{((q))}\right) + \rho \sum_{q=1}^{2} \mathrm{HSIC}(\boldsymbol{Q}_{\mathrm{C}}^{(q)}\boldsymbol{x}^{(q)}, \boldsymbol{Q}_{\mathrm{P}}^{(q)}\boldsymbol{x}^{(q)}), \tag{34}$$

where, first two term are adversarial loss for distribution matching. The constraint on (33d) and (33e) are enforced as $\mathcal{R}(\boldsymbol{Q}_{\mathrm{C}}^{(q)})$ and $\mathcal{R}(\boldsymbol{Q}_{\mathrm{C}}^{(q)})$ respectively, where $\mathcal{R}(\boldsymbol{Q}^{(q)}) = \|\boldsymbol{Q}^{(q)}\mathbb{E}[\boldsymbol{x}^{(q)}(\boldsymbol{x}^{(q)})^{\top}](\boldsymbol{Q}^{(q)})^{\top} - \boldsymbol{I}\|_{\mathrm{F}}^{2}$. The constraint on (33c) is realized with Hilbert-Schmidt Independence Criterion (HSIC) [75]. HSIC measures the independence between two distribution. So, we minimize HSIC between estimated shared component and estimated private component to promote independence between shared and private components.

We show that under some reasonable conditions the block $\boldsymbol{p}^{(q)}$ can also be learned up to a matrix multiplication:

**Theorem 6.** *Assume that the blocks $c$, $p^{(1)}$ and $p^{(2)}$ are statistically independent, i.e., $p(c, p^{(1)}, p^{(2)}) = p(c)p(p^{(1)})p(p^{(2)})$. Then, if one of the following holds:*

*(i) Assumption 1 and assumptions in Theorem 1 are satisfied, and (33) is solved yielding solutions $\widehat{Q}_C^{(q)}$ and $\widehat{Q}_P^{(q)}$*

*(ii) Assumption 2 is satisfied and has same mixing matrix $A^{(q)} = A$ and (33) with $Q_P^{(q)} = Q_P$ and $Q_C^{(q)} = Q_C$ is solved yielding $\widehat{Q}_C^{(q)}$ and $\widehat{Q}_P^{(q)}$ as the solutions.*

*(iii) Assumption 1 is satisfied and $d_C$ paired samples $(x_\ell^{(1)}, x_\ell^{(2)})$ are available (weak supervision), and denote $\widehat{Q}_C^{(q)}$ and $\widehat{Q}_P^{(q)}$ as the solutions after solving (33).*

*Then, we have $\widehat{Q}_C^{(q)} x^{(q)} = \Theta c$ and $\widehat{Q}_P^{(q)} x^{(q)} = \Xi^{(q)} p^{(q)}$, for some invertible $\Xi^{(q)}$ for all $q = 1, 2$.*

*Proof.* For each case in Theorem. 6 (i) - (iii), we can prove

$$\widehat{c}^{(q)} = \widehat{Q}_C^{(q)} x^{(q)} = \Theta c, \ q = 1, 2 \tag{35}$$

using Theorems 1-3. The proofs are referred to Appendix B-D.

Let us denote

$$\widehat{p}^{(q)} = \widehat{Q}_P^{(q)} x^{(q)} = \widehat{Q}_P^{(q)} A^{(q)} \begin{bmatrix} c \\ p^{(q)} \end{bmatrix} = H^{(q)} \begin{bmatrix} c \\ p^{(q)} \end{bmatrix}, \tag{36}$$

where $H^{(q)} = \widehat{Q}_P^{(q)} A^{(q)} \in \mathbb{R}^{d_P^{(q)} \times (d_C + d_P^{(q)})}$. Note that the constraint (33c) implies that the mutual information between $\widehat{p}^{(q)}$ and $\widehat{c}^{(q)}$ is zero, i.e.,

$$I(\widehat{p}^{(q)}; \widehat{c}^{(q)}) = 0.$$

Note that $\widehat{p}^{(q)} \to \widehat{c}^{(q)} \to \Theta^{-1} \widehat{c}^{(q)} = c$ is a Markov chain. This is because when conditioned on $\widehat{c}^{(q)}$, $\Theta^{-1} \widehat{c}^{(q)}$ becomes constant, making it independent of $\widehat{p}^{(q)}$. This allows us to use the data processing inequality [76, Theorem 2.8.1], which results in the following:

$$I(\widehat{p}^{(q)}; \widehat{c}^{(q)}) \geq I(\widehat{p}^{(q)}; \Theta^{-1} \widehat{c}^{(q)}) = I(\widehat{p}^{(q)}; c)).$$

Since mutual information is always non-negative, the above implies that $I(\widehat{p}^{(q)}; c) = 0$. This implies that $\widehat{p}^{(q)} = H^{(q)} \begin{bmatrix} c \\ p^{(q)} \end{bmatrix}$ is independent of $c$. Hence, $H^{(q)}[1 : d_C] = 0, \forall q$.

Therefore $\widehat{p}^{(q)} = H^{(q)}[d_C + 1 : d_C + d_P^{(q)}] p^{(q)} = \Xi^{(q)} p^{(q)}, \forall q$, where $\Xi^{(q)} = H^{(q)}[d_C + 1 : d_C + d_P^{(q)}]$. Note that $H^{(q)}$ is full row-rank because of constraint (33e). This implies that $\Xi^{(q)}, q = 1, 2$ are invertible matrices.

This concludes the proof.

$\square$

## H.1 Validation of Theorem 6

Fig. 7 presents numerical validation for Theorem 6.

**Hyperparameter Setting** The hyperparameter setting is the same as mentioned in Appendix. F. We solve (34) to obtain $\widehat{Q}_C^{(q)}$ and $\widehat{Q}_P^{(q)}$ to recover the shared and private components, respectively. For learning $Q_P^{(q)}$, we use Adam optimizer and set initial learning rates to be $0.001$. Also we set the regularization parameter $\omega = 10.0$ and $\rho = 50.0$.

**Data Generation:** We set $d_{\text{C}} = 2$ and $d_{\text{P}}^{(q)} = 1$ as in the previous synthetic experiments. We sample $\boldsymbol{c} \sim \text{VonMises}(2.5,\ 2.0)$ The private components are sampled from $p^{(1)} \sim \text{Beta}(1.0,\ 3.0)$ and $p^{(2)} \sim \text{Gamma}(0.5,\ 3.0)$ distributions. Each element of mixing matrices are sampled from $\boldsymbol{A}_{ij}^{(q)} \sim \mathcal{N}(0,1)$, $q = 1, 2$.

**Result:** Fig. 7 shows the result for proposed method for private component identification. The first column shows the data domain, the second column shows the true and extracted shared component, and the third and fourth columns shows the true and extracted private components. Especially, the last row of the third and fourth columns shows the plot of ground truth $\boldsymbol{p}^{(q)}$ on $x-$axis and $\widehat{\boldsymbol{p}}^{(q)}$ on the y-axis. The plot is approximately a straight line which indicates that the estimated private components $\widehat{\boldsymbol{p}}^{(q)}$ are scaled version (i.e., invertible linear transformations) of ground truth private components. This verifies our Theorem 6.

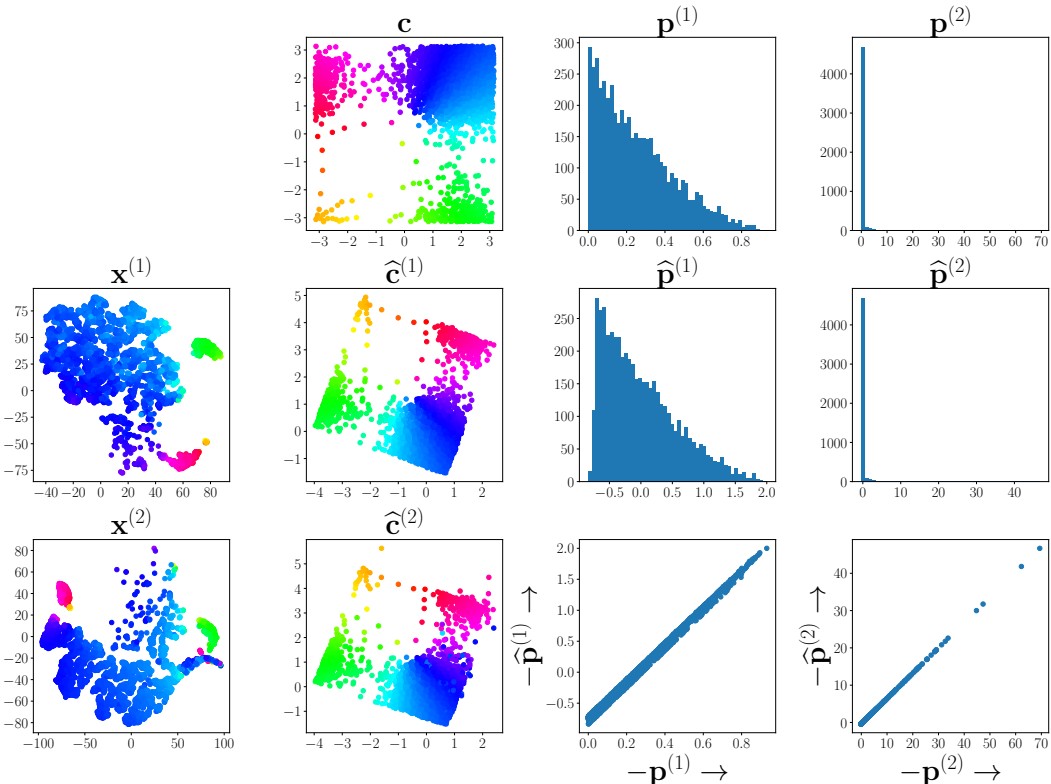

Figure 7: Validation of Theorem 6 $d_{\text{C}} = 2$ and $d_{\text{P}}^{(1)} = 1$.

