# OpenReview forum: "Identifiable Shared Component Analysis of Unpaired Multimodal Mixtures"
_NeurIPS.cc/2024/Conference — NeurIPS 2024 poster_

### Official Review · Reviewer_YjhB · 2024-07-08

**Soundness:** 3
**Presentation:** 3
**Contribution:** 2
**Rating:** 5
**Confidence:** 2

**Summary:**

This work presents a method for performing a shared component analysis (SCA) in the case of multi-modal unpaired data drawn from a linear mixture. This problem (and method) will be referred to as Unaligned SCA. Unaligned SCA is tackled by matching the probability distributions of the embedded (features) multi-modal data. More specifically, the authors draw inspiration from the traditional adversarial loss used in GANs, and formulate the component analysis algorithm as a min-max optimization problem where a discriminator (a neural network) is trained to maximize the confusion between the alignment of two representations (typically from two different modalities), and then the best alignment is sought in order to “fool” the current discriminator. The alignment matrices are structured so that the algorithm can distinguish between components shared across modalities and private components specific to each modality. The authors claim that the shared components can be identified up to the same ambiguities as those identifiable in the aligned case. Furthermore, the authors explain that while there are other methods that attempt to solve the Unaligned SCA, the conditions of the proposed algorithm are considerably milder.

The algorithm is then extended to cases where additional knowledge is present. First the algorithm is modified to accommodate the scenario where the data is generated by a single modality (uniform modality). Then, the algorithm is modified to accommodate the case where some data pairing is available (similar to a weakly supervised case). The authors show that by adding appropriate constraints  the shared component can be identified under milder conditions.

The author provide first a theoretical analysis with numerical simulations, and then some concrete applications of Unaligned SCA for the problem of domain adaptation (same modality), Multi-lingual Information Retrieval (only in the appendix, same modality) and Single Cell Sequence Analysis (multi modality with and without pairing).

**Strengths:**

- The work tackles an important area of research with potential applications that range from explainability, to SSL, or multi-modal problems.
- The method seems very flexible:
    - The method proposed can work for completely unpaired data but if some paring is available it can take advantage of such additional knowledge.
    - The method proposed is meant for multi-modal scenarios but it can also work in homogenous use cases.
- All parameters are well documented in the appendix and code was part of the additional material.

**Weaknesses:**

I would divide the weaknesses into two groups: the empirical evaluation, of which I am fairly confident about, and the theoretical analysis, of which due to my limited knowledge in this field I am less confident of. I will share here my concerns as solving them might also be helpful for other readers in my position

*Empirical Analysis.*

I find the empirical analysis weak. The multi-modal and unpaired scenario results are underwhelming, while I find the domain adaptation (same modality) potentially problematic due to the use of CLIP as pre-processing step, and the lack of a strong recent baseline (less problematic than the CLIP reason).
- The only practical results where the data are multi-modal and unpaired are the ones shown in Figure 4 first blue dot where the accuracy is about 10%. Since this is the only case presented it is unclear if the method, in practice, cannot cope in this scenario or if the specific problem chosen is particularly challenging (in which case other evidence would be maybe better).
- In the domain adaptation experiments the paper reads: “The images are pre-processed by the pretrained CLIP model [34] that uses ViT-L/14 transformer architecture.” It is not clear if ALL baselines used the CLIP embeddings as pre-processing, or if this was only done for the proposed algorithm. For fair comparison the same pre-processing should be applied to all algorithms.
- Additionally, CLIP is known to have been pre-trained on a large and diverse dataset and there is a good chance it has been trained on Home-Office and Office-31 too, so it is difficult to appreciate the ability of the proposed method when using such a powerful pre-processing step (which as I mentioned might have been trained on theses datasets, including the their test sets). So making sure the pre-processing is equal is necessary, I would also encourage to present results with a less powerful pre-processing step (e.g., something pre-trained on ImageNet either supervised or SSL style like SimCLR) in order to better distinguish the contribution of CLIP vs every baseline and the proposed model.
- The authors use a lot of baselines as comparison however all these baselines seem to be fairly old (all before 2020?). I would suggest comparing with a stronger baseline (either check the leaderboard or here are some suggestions [1-4], note that not all might be immediately applicable).

[1] D3GM (https://arxiv.org/pdf/2401.05465)

[2] CLUE (https://arxiv.org/abs/2010.08666)

[3] LAMDA (https://arxiv.org/abs/2208.06604)

[4] SDM (https://arxiv.org/abs/2203.05738)


*Theoretical analysis.*
I have struggled to follow the theoretical explanation. Specifically, I understand the rationale behind formulation in eq (7) but I would fully agree with it if the samples were paired. I do not understand how 6(b) holds for unpaired samples. I suppose this is the explanation currently presented before assumption 1 but even after reading it I was left with the same question.

**Questions:**

A satisfactory answers to these points could improve the "contribution" (and partially the "soundness") of the work.
- Was CLIP used as a pre-process step for all the baselines in the Domain Adaptation? If not, could the authors present those results by using the same pre-processing step? As I mentioned above ideally without using CLIP due to the potential contamination of the test sets.
- Could the author explain why 6(b) holds for unpaired samples?

Further addressing these less critical aspects could improve the "Presentation" score (and partially the contribution see first point)
- I believe the paper would be stronger with more recent baselines as suggested above.
- I find unclear how Fig.1 was created. More explanation would help the understanding.
- I find Fig 2 not clear: why are there c1 and c2 and only p1, whereas I was expecting p1 and p2 and a common (shared) c? Reading if further it might be that c1 and c2 are the two axes of the common space $c$. If this is the case I’d make sure to clarify it.
- Assumption 1 is not clear to me. Is this saying that both points y1 and y2 leave in the same sub-space within the span of $(c, p^{(1)}, p^{(2)})$?  If so in which way is this useful?
- The authors say the results are an average of 5 runs, which is great, but  the standard deviation should also be reported.
- I find this sentence confusing: "First, it is unclear if (6b) could disentangle c from p(q). In general, Q(q)x(q)
could still be a mixture of c and p(q) yet (6b) still holds (e.g., when both c and p(q) are Gaussian.)" first it says it is unclear if they can be disentangled, but the whole identifiably relies on the ability to disentangle them no?
- In a couple of places in the manuscript the authors refer to an experiments where all the results are in the appendix. I would add a brief summary of the results so the main paper is self-standing (this happens in More Synthetic-Data Validation and Application (iii)).
- In a few places $p^{(2)}$ appears without a closed bracket, i.e., $p^{(2}$.
- Sometimes the authors used the comma (,) to separate thousands but other time they didn’t. I would use a uniform notation.

**Limitations:**

Yes, the authors have identified and clearly stated three main limitations:
- The fact that the conditions presented are sufficient while the necessary are not yet known.
- The fact that the method works only for linear mixtures (which limits the expressivity).
- Lastly, the fact that the theoretical derivation assume infinite data.

---

> ### Author Rebuttal · Authors · 2024-08-07
>
> **[Empirical Results, CLIP, and Recent Baselines]**
>
> **(i) “Only Practical Result is in Fig 4.”** We believe that there might have been some misunderstanding. Fig. 4 is used to validate Theorem 3. The blue dot markers suggest that conditions (a-b) in Theorem 1 might not be satisfied.
>
> To clarify, the paper studied three settings, corresponding to Theorems 1, 2 and 3. The motivation for studying Theorems 2-3 is that condition (a-b) in Theorem 1 seemed to be restrictive for many applications. Theorems 2 and 3 also work under **practical settings**, but with more structural info. relative to Theorem 1. Theorem 2 uses homogeneity and Theorem 3 uses weak supervision; see similar settings in [R17,R20,R22]. Theorem 1 is validated in Fig. 3; Theorem 2 is validated in Sec. 6, applications (i) and (iii), and Theorem 3 is validated in Fig. 5 and Sec. 6 application (ii).
>
> **(ii) Avoid Using CLIP and More Recent Baselines.** Thanks for this comment. The reviewer has a good point.
>
> We agree with the reviewer that the CLIP was trained on a large and diverse dataset and could have included Office31 and Office Home as well. Hence, it may not be fair to run the methods over the CLIP-learned space.  To address this issue, we follow the reviewer’s suggestion and use features from Resnet50 pre-trained over the ImageNet1k dataset. All methods use the same pre-trained features for a fair comparison.
>
> As per reviewer’s suggestion we also added two recent baselines:
>
> **ELS [R24]**  (ICLR 2023)
>
> **SDAT [R23]**  (ICML 2022)
>
> Note that we didn't use the reviewer-suggested active learning baselines as they applied to different settings.
>
> Please see Table 1 and 2 in the **attached PDF** for the new experiments.
>
> **[Regarding Distribution Matching (Eq(7) and Eq(6b))]**
> The first two terms (GAN loss) in Eq. (7) is used to enforce (6b), i.e., matching the distributions of random variables $\bf{Q}^{(1)}\bf{x}^{(1)}$ with $\bf{Q}^{(2)}\bf{x}^{(2)}$ [R25].
> We do not need paired samples to match the distributions. Intuitively, (7) “learns” $\bf{Q}^{(q)}, q=1,2$, in such a way that the samples of $\bf{Q}^{(1)}\bf{x}^{(1)}$ cannot be distinguished from the samples of $\bf{Q}^{(2)}\bf{x}^{(2)}$ in terms of distribution.
>
> **[Questions Regarding Contribution and Soundness]**
>
> **a.** CLIP: The reviewer has a great comment. No, the CLIP was not used for other methods. We agree with the reviewer that this should be rectified. We followed the reviewer’s suggestion and ran all methods using CLIP features (by throwing away the baselines’ original encoders). It turned out that all methods perform similarly well (**see Table 3 in the attached PDF**). As the reviewer pointed out, this is perhaps because CLIP memorizes too much information.
>
> **Tables 1 and 2** in the attached PDF show new results where every baseline uses features from ImageNet1k-pretrained ResNet50. We hope these new results would alleviate this concern.
>
> **b.** Please refer to **[Regarding Distribution Matching (Eq(7) and Eq(6b))].**
>
> **[Questions Regarding Presentation]**
>
> **a.** Please refer to **[Empirical Results, CLIP, and Recent Baselines]**
>
> **b.** To create Fig. 1, we first sample $\bf{c} \sim \cal{N}(\bf{0}, \bf{I}), \bf{c} \in R^2$, and set $\bf{\Theta}^{(q)}, q=1,2$ to two different rotation matrices. A unique color is picked for each sample of $\bf{c}$ and used to color both $\bf{\Theta}^{(q)} \bf{c}, q=1,2$. The purpose of Fig. 1 is to illustrate that $ \bf{\Theta}^{(1)} \bf{c} \stackrel{(d)}{=} \bf{\Theta}^{(2)} \bf{c}$ even when $\bf{\Theta}^{(1)}  \neq \bf{\Theta}^{(2)} $, which is clearly the case. We will make Fig.1 caption clearer.
>
> **c. [Clarification of Assumption 1]**
>
> (A1) aims to provide a way to characterize how different the latent distributions $P_{\bf{c},\bf{p}^{(1)}}$ and $P_{\bf{c},\bf{p}^{(2)}}$ are.
>
> Let $S^{(q)}$ denote the set of all possible ``stripes'' $\cal{A}^{(q)}$ described in (A1) for domain $q=1,2$. Then, (A1) simply says that there should not exist any two stripes $\cal{A}^{(q)} \in S^{(q)}, q=1,2$ such that the two latent distributions assign the same probability to all scaled versions of stripes $\cal{A}^{(q)}$. This makes the two distributions sufficiently different from each other, needed to identify common and private information in the two domains.
>
> Regarding $\bf{y}_1, \bf{y}\_2$: No, it is not saying that $\bf{y}\_1$ and $\bf{y}\_2$ live in the same subspace within the span of $(\bf{c},\bf{p}^{(1)},\bf{p}^{(2)})$. The vectors $\bf{y}_i^{(q)}$ are any linearly independent vectors in $\bf{R}^{d\_C+d\_P^{(q)}}$, whereas $\bf{c}$ and $\bf{p}^{(q)}$ are random vectors in $\bf{R}^{d\_C}$ and $\bf{R}^{d\_P^{(q)}}$, respectively.
>
> The **use** of the vectors $\bf{y}\_1, \bf{y}\_2, \dots$ is to characterize the stripes $\cal{A}^{(q)}$, which are ultimately used to characterize the difference between the two latent distributions,  $P\_{\bf{c},\bf{p}^{(1)}}$ and $P\_{\bf{c},\bf{p}^{(2)}}$.
>
> **d.** Due to time constraints in the rebuttal process, we could only produce the standard deviation for a couple of transfer tasks in the new experiments. Table 2 (first 2 rows) shows the corresponding **mean and standard deviations** for (Ar $\to$ Cl, and Ar $\to$ Pr) using 5 trials. We will include the new experiments with multiple runs and standard deviation in the revised version.
>
> **e.** Yes, identifiability implies disentanglement of $\bf{c}$ and $\bf{p}^{(q)}$. The two sentences highlight the challenge of ensuring this disentanglement by solving Problem (6), and provide an example where disentanglement fails, such as when $\bf{c}$ and $\bf{p}^{(q)}$ are Gaussian. This indicates that disentanglement is not always possible.
> The purpose of our work is to derive precise conditions under which disentanglement can be ensured.
>
> **f,g,h.** Thanks. We will fix the typos, and add a brief summary of the results in the main paper.
>
> For references, please refer to the **References** section in the **Overall Response**.

---

> > ### Comment · Reviewer_YjhB · 2024-08-08
> >
> > Thanks to these authors for their answers. I appreciate the explanations and the effort in running the domain adaptation experiments using my suggestions. I suggest using the ImageNet features as the main result in the paper and provide the CLIP ones in the Appendix (as it shows that a powerful feature extractor such as CLIP reduces most of the differences among all the algorithms).
> >
> > There is still one aspect I do not fully understand. Could the authors state explicitly which of the three experiment settings (i), (ii) or (iii) shown in the paper are at the same time multi-modal and unpaired?

---

> > > ### Author Response · Authors · 2024-08-08
> > >
> > > (1) We will follow the reviewer’s suggestion and use ImageNet features as the main result in the paper, while moving updated CLIP experiments to the Appendix.
> > >
> > > (2) All 3 applications (i.e., (i) image domain adaptation, (ii) single-cell sequence alignment, and (iii) multilingual embedding retrieval) are multimodal and unpaired. But their detailed settings vary. We proposed our unaligned SCA approach under three different settings, depending on how much structural information can be exploited.
> > >
> > > **[Setting 1] Multimodal and unpaired:** The setting uses $\bf{x}^{(i)} = \bf{A}^{(i)} \bf{z}^{(i)}$ where $\bf{z}^{(i)}=(\bf{c},\bf{p}^{(i)})$ for modality $i$. The different $\bf{A}^{(i)}$’s and $\bf{p}^{(i)}$’s both represent the modality discrepancies. Synthetic data was used to validate the setting. We argued the condition needed here for identifiability was too strong for many applications. This was the motivation for us to consider Settings 2-3.  (see Sec. 4 Line 203-208).
> > >
> > > **[Setting 2] Multimodal and unpaired; the modalities share a homogeneous feature space (sometimes called multi-domain setting):** The setting uses $\bf{x}^{(i)} = \bf{A} \bf{z}^{(i)}$ where $\bf{z}^{(i)}=(\bf{c},\bf{p}^{(i)})$ for modality $i$. The modality/domain differences are captured by $\bf{p}^{(i)}$’s. Unlike Setting 1 where $\bf{A}^{(i)}$ varies across $i$, here the mixing systems $\bf{A}^{(i)}=\bf{A}$ for all $i$. This is often called a homogeneous multi-domain setting, which is a more special case of multimodal learning. This setting makes sense when the data $\bf{x}^{(i)}$ for all domains share the same feature space, often used in applications like image-to-image domain adaptation [R17] and image-to-image style translation [R21]. **Applications (i) and (iii) were used to validate the method under this setting**. Application (i) is on domain adaptation of images. The images from different domains are unpaired. Application (iii) is Multilingual retrieval problem. The modalities correspond to unpaired words in different languages.
> > >
> > >
> > > **[Setting 3] Multimodal and largely unpaired (with a small number of paired data):** The setting uses $\bf{x}^{(i)} = \bf{A}^{(i)} \bf{z}^{(i)}$ where $\bf{z}^{(i)}=(\bf{c},\bf{p}^{(i)})$ for modality $i$. The vast majority of data are unpaired. But there exists a small number of paired data (for example, in our experiment of Fig 4, the total number of data in each domain is 1,874. We considered cases where 0 to 256 paired data exist, i.e., 13% at its maximum). This setting is considered realistic in applications such as [R22, R26]. **Application (ii) was tested under this setting**. It corresponds to the single-cell experiment. Modalities correspond to unpaired RNA sequences and ATAC sequences.
> > >
> > > [R26] *Wang et. al., 2020. Semi-supervised Learning for Few-shot Image-to-Image Translation.*

---

> ### Comment · Reviewer_YjhB · 2024-08-09
>
> Thank you for your explanation. I now think I understand where the misunderstanding comes from: by multi-modal I was expecting two completely different data modality (e.g., image and sound). By this definition I thought the only true multi-modal setting was application (ii) with one type of data being RNA sequences and one being ATAC sequences. After a bit of further investigation I think even this setting is not quite multi-modal (as in two separate and different modality) as I believe (but I am not a biologist) both sequences are actually made by the 4 DNA bases. Similarly, the other two applications are mono-modal: application (i) is images (taken from different type of cameras) and application (iii) is text (coming from different languages).
>
> I appreciate the in all three applications the data have different distributions, but I think it should be made clear that the focus is not multi-modal settings (as in multi-modality type of data) but rather in same modality with a distribution shift, also known as domain adaptation. I believe changing the title and the explanation to domain adaptation rather than multi-modal would help setting the right expectation for the reader. I would also encourage the authors to provide a short summary of the results of application (iii) rather than delegating everything to the appendix.
>
> With all the additional experiments and clarifications provided during this rebuttal I believe the authors have increased the quality of their work to the following:
>
> Soundness: 3: Fair -> Good
>
> Presentation: 3: Fair -> Good
>
> Contribution: 2: Fair
>
>
> The reason for keeping the contribution as Fair is that:
> - The method was tested only on domain adaptation experiments rather than what I thought was a multi-modality setting.
> - It was shown in the additional experiments that using a powerful feature extractor (CLIP in this case) is arguably more beneficial than any sophisticated algorithm.
>
> I am ok to increase my overall recommendation from 4 to 5.

---

> > ### Author Response · Authors · 2024-08-09
> >
> > We thank the reviewer for their detailed and constructive discussion during the rebuttal.
> >
> > The comment about multi-modality seems to be a terminology issue. We will follow the reviewer's suggestion and change the term to "multi-domain".
> >
> > We will also add the summary of results of application (iii) in the main paper.
> >
> > Nonetheless, the terminology issue doesn’t seem to affect our contributions. We believe that our contribution lies in providing rigorous understanding of the proposed unaligned multi-domain problem structure. Our synthetic and real-data experiments were designed to validate that understanding. The unaligned SCA problem is of great interest as a latent component analysis problem, like ICA, PCA and CCA. Its identifiability has been elusive and our work filled this gap.
> > We wonder if the reviewer could re-assess the contribution from the identifiability research viewpoint, rather than the “multi-modality” vs “mono-modality” viewpoint.
> >
> > In any case, we sincerely thank the reviewer for the comments and discussion, and for pushing us to improve our experiments and presentation.

---

### Official Review · Reviewer_tDhK · 2024-07-13

**Soundness:** 2
**Presentation:** 3
**Contribution:** 2
**Rating:** 5
**Confidence:** 3

**Summary:**

The paper considers the identifiability of shared components from a linear mixture. The theory requires multiple domains. However, compared to previous works, the required domains do not need to be aligned in this work. A practical estimation model has been proposed according to the theory.

**Strengths:**

1. The discussion on the related work is comprehensive.

2. The experiments have been conducted on both synthetic and real-world datasets.

3. The proposed algorithm looks pretty neat.

4. Limitations have been discussed in detail together with potential next steps.

**Weaknesses:**

1. Since there are already many works in learning shared components in the nonlinear setting, and some of them can even handle unpaired mixtures, the linear setting appears less appealing in comparison.

2. Assumption 1 is similar to the one used in the previous works, which should be highlighted earlier in the paper.

3. The discussion of the assumption of hyper-rectangle support is missing. Is it restrictive? Maybe some real-world examples could be helpful.

**Questions:**

1. I didn't fully understand the proof in Line 780--does the usage of data processing inequality require a Markov chain? If so, has it been shown in the proof?

2. Could you please elaborate more on the connection between the proposed theory and previous work focusing on identifying content and style variables? It seems like they share a similar goal.

**Limitations:**

The authors have discussed the limitations.

---

> ### Author Rebuttal · Authors · 2024-08-07
>
> **[Regarding "Many Shared Component Learning Methods Exist"]** We would like to note that the existing **identifiability research** on shared component learning (including nonlinear mixture based ones) from **unaligned** multi-domain data is in fact rather limited (although empirical studies are abundant). In addition, the limited existing identifiability results cannot cover our settings, and thus study under our settings is still of its own significance.  We reviewed these identifiability studies in the manuscript. We reiterate with a bit more details as below:
>
> The work in [R16] considers identifiability of unaligned data learning using linear transformation. But they did not model shared and private latent components. Their identifiability conditions are also much more restrictive relative to ours (e.g., they require the 3rd-order moments of data are high-rank tensors, which is hard to satisfy and a little counter-intuitive).
>
> The work of [R17,R20] considered a non-linear mixing function for the homogeneous data case. Compared to our work, [R17,R20] required at least a large number (i.e., $ 2 d_P + 1$) of domains, component-wise independence of latent variables, and much more stringent domain variability assumptions as discussed in Line 167-173 of the manuscript. Our method can work with as few as only 2 domains and the domain variability condition (A1) that we use is much more relaxed.
>
> As detailed from Line 105-118, the work [R19] considered the same generative model as ours, but operated under much more stringent assumptions on the latent variables, such as all variables being component-wise independent and having unit variance.  Both conditions are not needed under our framework.
>
> The works [R2,R14,R18] all consider identifiability of the shared components under linear or nonlinear mixture settings. However, they all assume that the cross-domain data are aligned, which is significantly simpler than our setting.
>
> **[Regarding the Rectangle Assumption]** We consider the assumption of hyper-rectangle support to be not very restrictive. Any collection of real-valued features could form a hyper-rectangle. For example, in image-to-image translation between animal faces (e.g., dog and cat images be the two domains) [R21], the position and orientation of the animal is generally the shared information whereas the appearance of the animal is the private information. If these two aspects are represented by two real-valued components in the latent space, then their supports could easily form a rectangle.
>
> **[Questions]**
>
> **Q1.** Yes, the data processing inequality requires a Markov chain. And we do have a Markov chain: $\hat{\bf{p}}^{(q)} \to \hat{\bf{c}}^{(q)} \to \bf{\Theta}^{-1} \hat{\bf{c}}^{(q)} = \bf{c} $. It is a Markov chain because conditioned on $\hat{\bf{c}}^{(q)}$, $\bf{\Theta}^{-1} \hat{\bf{c}}^{(q)}$ is a constant, and thus independent of $\hat{\bf{p}}^{(q)}$.
> It was not explicitly written in the manuscript. We will add this clarification.
>
> **Q2.** Please refer to **[Regarding “Many Shared Component Learning Methods Exist”]**
>
> &nbsp;
>
> ## References
>
> [R2] Lyu et al., 2022. Understanding Latent Correlation-Based Multiview Learning and Self-Supervision: An Identifiability Perspective.
>
> [R14] Kugelgen et al., 2021. Self-Supervised Learning with Data Augmentations Provably Isolates Content from Style.
>
> [R16] Gulrajani et al., 2022. Identifiability Conditions for Domain Adaptation.
>
> [R17] Xie et al., 2023. Multi-Domain Image Generation And Translation With Identifiability Guarantees.
>
> [R18] Sorensen et al., 2021. Generalized Canonical Correlation Analysis: A Subspace Intersection Approach.
>
> [R19] Sturma et al., 2023. Unpaired Multi-Domain Causal Representation Learning.
>
> [R20] Kong et al., 2022. Partial disentanglement for domain adaptation.
>
> [R21] Choi et al., 2019. StarGAN v2: Diverse Image Synthesis for Multiple Domains.

---

> > ### Comment · Reviewer_tDhK · 2024-08-13
> >
> > Thank you for your response. I would like to maintain my positive score.

---

### Official Review · Reviewer_1Nhs · 2024-07-14

**Soundness:** 2
**Presentation:** 2
**Contribution:** 2
**Rating:** 6
**Confidence:** 3

**Summary:**

This work considers a problem similar to classical Canonical Correlation Analysis (CCS), which assumes a linear generative model for data $(x_1, x_2)$: $x_1=W_1z$, $x_2=W_2z$ and aims to identify the underlying components.

This problem has been extended previously to include "private information": $x_1=W_1z_1, z_1=[c,p_1]$, $x_2=W_2z_2, z_2 = [c,p_2]$ for common $c$ and independent $p_j$.

The current work further assumes that data is *unpaired* and rather than mapping each pair ($x_1$, $x_2$) such that $z_1$, $z_2$ are close together (in some metric), it is proposed that all $x_1$ are mapped to be similar *in distribution* to the mapped $x_2$s.

**Strengths:**

The paper aims to provide rigorous criteria in which underlying generative factors are identifiable in the extended CCA problem it tackles (unpaired CCA with private information).

The results show improvement over benchmarks indicating promise to the approach.

**Weaknesses:**

High level:
* While I understand the basics I am not an expert in the area of CCA, but I find the paper fairly difficult to follow. More explanation would be helpful, e.g.
    - [28] why is any linear mixture model ill-posed (is that strictly true in *every* linear mixture case?)
    - [47] what is meant by "facilitating one-to-many translations", the context/meaning is unclear.
* The theoretical part of the paper relates to a simple linear model, but none of the experiments follow this model
    - e.g. the algorithm is applied to CLIP embeddings, which are not "the data", so to make claims about a simple linear model z=Ax and then apply it to CLIP seems incongruous. This experiments seem to relate more to a CCA-based "loss function" that takes representations and looks to align them/encourage independent factors etc.
    - other experiments appear to be on discrete data, which the methods doesn't apply to, presumably these are also represented as some intermediate step?
    - it seems strange to propose a simple linear model, present theoretical results about identifiability that rely on that simplicity but make a dramatic departure in the experiments where the assumptions clearly do not hold and the notion of identifiability is unclear.
*  if the work does achieve an improvement in a CCA type setting, it seems appropriate to compare with other CCA methods on suitable data. Adding results on more complex data/representations may be of additional interest.

Assumption 1
 - hard to parse and could be made more clear.
 - unclear if correctly defined, don't vectors y need to be orthogonal to subspace P? It seems extremely loose to the point of simply saying $P_{c,p_1} \ne P_{c,p_2}$ (specifying where any difference lies to this might add clarity).

Theorem 1
 - is this saying that if all dims of c are distributed differently, p(z)'s can only match (e.g. under GAN loss) by correctly aligning each dim? If so, that is pretty intuitive and it could be made clearer that you are putting that mathematically and proving it for the sake of rigour.
 - I have not been through the proof, 5+ pages of proof without a sketch in the paper might be more suitable for a journal as appendices are not typically expected to be reviewed in detail.

Overall, there may be useful results in the paper, but in my view it should be re-written to make more clear what it is doing. It seems a confusing mix of simple linear generative model and related CCA methodology mixed with much more complex representations (e.g. CLIP) passed through a GAN + linear layer.

**Questions:**

see weaknesses

**Limitations:**

see weaknesses

---

> ### Author Rebuttal · Authors · 2024-08-07
>
> **[Linear mixture models (LMMs) are Ill-posed]**
>
> In general, LMMs are not identifiable. Because for any $\bf{y}=\bf{A}\bf{x}$, where both $\bf{A}$ and $\bf{x}$ are unknown, one can find an infinite number of invertible $\bf{Q}$ such that $\bf{y}=\bf{AQQ}^{-1}\bf{x}$. Then, both $(\bf{A},\bf{x})$ and $(\bf{AQ},\bf{Q}^{-1}\bf{x})$ are equally fit to the data $\bf{y}$, making the problem ill-posed in terms of identifiability (or, solution uniqueness) [R3-R5]. In our case, we aim to identify two blocks in $\bf{x}$, i.e., $\bf{x}=[\bf{c},\bf{p}]$. The same ill-posedness remains.
>
> **[One-to-many translations]**
>
> Translation means changing the appearance of a sample in $\bf{x}^{(1)}=\bf{A}^{(1)}[\bf{c},\bf{p}^{(1)}]^T$ to its corresponding samples in the other domain. Note that the content is given by $\bf{c}$, and the appearances are controlled by $\bf{A}^{(q)}$ and $\bf{p}^{(q)}$. One-to-many translation means that one can combine the $\bf{c}$ extracted from $\bf{x}^{(1)}$ with many different $\bf{p}^{(2)}$; see examples in [R6].
>
> **[Using CLIP as Pre-processing]**
>
> Please note that we only used CLIP as pre-processing (analogous to using PCA in the old days). Powerful pre-processing tools like CLIP and pretrained vision models (eg. ResNet) can map images to approximately linear subspaces [R7] (also see linear probing experiments in [R8-R9]).
>
> **[Theory-Experiment Consistency]**
>
> We respectfully disagree with the comment that our experiments are "dramatic departure" from our theory. Please note that our experiments in Fig. 3, 5, and 7 **exactly follow the LMM**. The experiments on images, single-cell data, and language data used proper pre-processing in order to map data to linear subspaces, approximating our model and supporting the * **usefulness** * of the model in practice.
>
> **[“Discrete Data”]**
>
> No. We did not run experiments with discrete data. We applied our methods on the continuous feature representation space of the data. Those features were obtained from gene expression counts [R10] and fast-Text embeddings [R11] in single cell and multi-lingual experiments, respectively. Modeling such features as continuous random variables is a common practice [R12,R13].
>
> **[The Significance of Understanding the LMMs]**
>
> Understanding the LMM-based unaligned SCA is the first step towards more complex models. This is how studies evolved from ICA [R3], PCA, and NMF [R4] (all LMMs) to provable nonlinear mixture models. However, there has been no existing theoretical support for the identifiability of the clearly important unaligned SCA model considered in this work.
>
> **[Comparing with CCA]**
>
> Let us clarify: CCA cannot work under our setting. The limitation of CCA is that CCA needs to know the one-to-one correspondence between multi-domain data. For example, for multilingual word alignment, it needs paired data $(\bf{x}^{(1)}\_\ell,\bf{x}^{(2)}\_\ell )\_{\ell=1}^L$ where $\bf{x}^{(q)}\_\ell,$ $q=1,2$ represent the same entity (e.g., "cat") in two languages. In our case, the data is not paired, i.e., $\bf{x}^{(q)}\_\ell,$ $q=1,2$ need not correspond to the same word. Hence, our method is called "unaligned SCA", but CCA is essentially "aligned SCA". Their settings are fundamentally different.
>
> **[Clarifying Assumption 1 (A1) ]**
>
> (A1) aims to provide a way to characterize how different the latent distributions $P_{\bf{c},\bf{p}^{(1)} }$ and $P_{\bf{c},\bf{p}^{(2)}}$ are.
>
> Let $S^{(q)}$ denote the set of all possible "stripes" $\cal{A}^{(q)}$ described in (A1) for domain $q=1,2$. Then, (A1) simply says that there should not exist any two stripes $\cal{A}^{(q)}$$\in S^{(q)},~q=1,2$ such that the two latent distributions assign the same probability to all scaled versions of stripes $\cal{A}^{(q)}$. This makes the two distributions sufficiently different from each other.
>
> **(i) Vectors $y$ orthogonal to subspace $\cal{P}$**
>
> No, $\bf{y} \perp \cal{P}$ is not needed since this orthogonality constraint leaves $S^{(q)}$ unchanged, i.e., $\hat{S}^{(q)}= \\{\cal{A}^{(q)}\in S^{(q)}|~\hat{\bf{y}}_i^{(q)}\perp\cal{P}^{(q)}\\}=S^{(q)}.$ To see this, first it is clear that $\hat{S}^{(q)}\subseteq S^{(q)}$. Second, any $\cal{A}^{(q)}\in S^{(q)}$ is equal to some $\hat{A}^{(q)}\in\hat{S}^{(q)}$ constructed using $ \bf{\Pi}\_{\cal{P}^{(q)}}\{y}\_i^{(q)}$ instead of  $\bf{y}\_i^{(q)}$, i.e., the projection of $\bf{y}\_i^{(q)}$ onto the orthogonal subspace of $\cal{P}^{(q)}$. Hence $S^{(q)}\subseteq \hat{S}^{(q)}$, and thus $S^{(q)}=\hat{S}^{(q)}$.
>
> **(ii) Difference between (A1) and $P_{\bf{c},\bf{p}^{(1)}}\neq P_{\bf{c},\bf{p}^{(2)}}$**
>
> We are a little confused by the reviewer’s comment that "It seems extremely loose … saying $P_{\bf{c},\bf{p}^{(1)}}\neq P_{\bf{c},\bf{p}^{(2)}}$". **We hope to clarify that we never used $P_{\bf{c},\bf{p}^{(1)}}\neq P_{\bf{c},\bf{p}^{(2)}}$ but only (A1).** For $P_{\bf{c},\bf{p}^{(1)}}\neq P_{\bf{c},\bf{p}^{(2)}}$ to hold, the two joint PDFs can be exactly the same everywhere except for an arbitrarily small subset of their domain. However, Assumption 1 only holds if the two joint PDFs are sufficiently different, by comparing the measures over the specifically defined “stripe” regions.
>
> **[Theorem 1 clarification]**
>
> **(i)  Meaning of theorem 1**, the reviewer’s description is not entirely accurate, but partially correct. Theorem 1 (a) states that if all dimensions of $\bf{c}$ are distributed differently (as in Line 179) and independent, then $p(\bf{z}^{(1)})$ can only match $p(\bf{z}^{(2)})$, where $\bf{z}^{(q)}=\hat{\bf{Q}}^{(q)}\bf{x}^{(q)}$, only when $\bf{z}^{(1)}$ and $\bf{z}^{(2)}$ are aligned, i.e., $\bf{z}^{(1)}=\bf{z}^{(2)}=\bf{\Theta}\bf{c}$. This is called block-identifiability [R14,R15].
>
> **(ii) Regarding the proof length**, We will include a proof sketch in the main paper when it fits; otherwise, we will include it in the appendix with clear pointers in the main paper.
>
> &nbsp;
>
> For references, please refer to the **References** section in the **Overall Response.**

---

> ### Comment · Reviewer_1Nhs · 2024-08-12
> **Reviewer response**
>
> I realise (as an author) that reviews can sounds attacking. I would like to stress, since your response doesn't seem to acknowledge any change, that I appreciate this line of work and my comments are to improve the paper if possible.
>
> * **LMMs & 1-many translations**: these are points of clarity to "the general reader", not just me, I think the paper could be more clear and standalone than it currently is
> * **Preprocessing**: you mention "linear subspaces", I'm not sure I follow, for sure various representation models give representations that already untangle much complexity in the data (e.g. so that semantically similar items are clustered). CCA acts on the raw data, you are acting on representations. It does not make the approach invalid, but it should be more clearly stated that is what you are doing. You are in effect heavily relying on what other models achieve, which is completely arbitrary with respect to your contribution. In effect you are providing a loss function to wrap around pre-trained representations along the principles of CCA. This is not what is in the abstract for example: "This work takes a step further, investigating shared component
> identifiability from multi-modal linear mixtures where cross-modality samples are unaligned". Given you don't know what the "representation model" has done, it detracts from identifiability claims, which typically refer to the data itself and should at least be caveated.
> * **Data**: you do run experiments on discrete data: text is discrete. As above you rely on representations that have already done a lot of work in re-representing it. It would be better, in my view, to demonstrate that the linear CCA-type workings actually work as expected on multiple appropriate (simpler) datasets and then show that that still holds for more complex scenarios where non-linear encoders (or similar) have effectively taken the non-linearity into account. Identifiability should relate to factors that those underlying models have identified.
> * **Thm 1**: I think an intuitive explanation in the paper would improve readability/understandability.

---

> > ### Author Response · Authors · 2024-08-13
> >
> > We would like to stress that we absolutely found the reviewer’s comments valuable for improving the paper’s clarity. It was our fault that we missed adding sentences that commit changes (while concentrating too much on the 6000 character limitation), which was not our intention. We do agree with the reviewer: any suggestion that may help the general readers to better understand the paper is appreciated. We thank the reviewer for the help on clarity and will definitely make revisions accordingly.
> >
> > **[LMM and 1-many translations]**
> >
> > We agree with the reviewer that explanations to these points could make our paper more self-contained. Hence, we will add the explanations (provided in the rebuttal) in a separate "Preliminaries" section in the Appendix with clear pointers in the main paper.
> >
> > **[Preprocessing]**
> >
> > By "linear subspaces", we mean representation spaces where the representations are likely to be linear mixtures of semantic information. As mentioned in our original rebuttal (**[Using CLIP as Pre-processing]**), this has been observed to be the case for embedding spaces of neural networks such as CLIP [R7], word embeddings [R30] etc.
> >
> > We would like to clarify that our method is always applicable wherever CCA is applicable, since CCA shares the same generative model [R18, R29] (also see section 2 Aligned SCA in the manuscript) as ours (the only difference is that CCA further requires the cross-modality samples to be aligned according to their content). Note that CCA also uses pre-processed features representations for complex real-world data (e.g., image, text) [R28, R18]. This is because these complex real-world data might not follow the linear mixture model in Eq. (1) in the manuscript, however the pre-processed representations might. Note that it is common for identifiability works to use pre-processed features for real-data validation of their Theorems [R18]. However, we understand the reviewer’s comment on the applicability to complex data directly. We will explain in more detail in the beginning of the experiment section regarding why preprocessing is involved.
> >
> > We also hope to remark that the sentence in our abstract "*This work takes a step further, investigating shared component identifiability from multi-modal linear mixtures where cross-modality samples are unaligned*" is an accurate claim. Note that our claim is for multi-modal **linear** mixtures. Therefore, for complex datasets, it is necessary to find appropriate linear representation spaces. We will add more clarifications/reminders when it comes to the experiment section.
> >
> > **[Data]**
> >
> > We agree with the suggestion of first using simpler raw data and then representations of more complex data to run experiments. The presented experiments in fact may have implicitly reflected this comment.
> >
> > To explain, note that the single-cell data is not pre-processed using any encoder but a normalized (zero mean and unit std) version of the raw-data, which is a simpler dataset as the reviewer mentioned.  The more complex image and language data were preprocessed by existing encoders.
> >
> > Following the reviewer’s suggestion on “simpler datasets ---> harder datasets” comment, we will change the order of presenting the single-cell experiment and the other experiments, to make this more explicit.
> >
> > **[Thm 1]**
> >
> > Thank you for your suggestion. We will include a simpler, intuitive explanation of Theorem 1 in the revised version.
> >
> > **References**
> >
> > [R28] Shi et al., 2019. Image Retrieval via Canonical Correlation Analysis.
> >
> > [R29] Ibrahim et al., 2020. Reliable Detection of Unknown Cell-Edge Users via Canonical Correlation Analysis.
> >
> > [R30] Mikolov et al., 2013. Efficient Estimation of Word Representations in Vector Space.

---

> > > ### Comment · Reviewer_1Nhs · 2024-08-13
> > > **thanks for the response**
> > >
> > > Thanks, if the proposed changes are made I think the readability and understanding of the paper should improve. I have not been through the proof in detail, but the approach makes sense and the empirical results suggest the method works. Assuming the proposed changes are made, which are not substantive, I recommend the work being published.
> > >
> > > Score 4 --> 6

---

> > > > ### Author Response · Authors · 2024-08-13
> > > >
> > > > We would like to thank the reviewer for helping us improve the clarity and for the constructive communication.

---

### Official Review · Reviewer_DxMx · 2024-07-23

**Soundness:** 4
**Presentation:** 3
**Contribution:** 3
**Rating:** 7
**Confidence:** 3

**Summary:**

This work considers the identifiability of linear latent representations that are shared (i.e., identical) across data modalities, in the special case that they are unaligned/unpaired.
The approach leverages GAN-style training to achieve divergence minimization between the latent distribution of each modality.
The approach appears to be restricted to two modalities.
Under the assumption of shared latents, sufficient (not necessary) conditions for identifiability are presented, which are milder and, thus, more general than existing studies.
Further, structural constraints based on side information are introduced to further relax identifiability conditions.
Several experiments on simulation and real-world data are provided.

**Strengths:**

Originality :
- The work introduces a combination of novel and known ideas in a clever formulation that yields new, less restrictive conditions for identifiability of shared signals from two modalities.
- The work differs from and extends previous contributions, dealing two unaligned/unpaired modalities.
- The work further relaxes the identifiability conditions via structural constraints based on additional side information that may be available in certain problems.
- The manuscript cites related work on identifiability conditions for aligned/paired data, as well as unaligned/unpaired results using stricter ICA conditions, also linking the work to nonlinear studies, adequately indicating the sources of inspiration.
Quality :
 - The work is technically sound, including proofs for identifiability claims.
 - The theoretical claims are well supported by the experiments.
Clarity :
 - The work is well written and organized, focusing on the key points and contributions.
Significance :
 - The results appear to be quite meaningful, with a potentially wide range of application.

**Weaknesses:**

Quality :
 - The code is not too friendly to readers, lacking higher 1-to-1 correspondence with the notation in the paper. Suggest improving documentation and tidying up the codebase for readability.
 - Figure 5 does not seem to replicate well in `synthetic_train.ipynb` --> Clarify
 - The numerical validations (simulation) are limited to 100,000 samples. Could you illustrate performance at 10,000, at 1,000, and at 100 samples? Many multimodal applications are limited to sample-poor regimes (N < 100), where classical CCA is one of the few performant methods, so it would be useful to assess the performance of unaligned SCA at varying sample sizes, and perhaps include comparable CCA results. Is there a summary measure (like Amari distance) that could be reported alongside figures?
Clarity :
 - The paper contains some typos that limit the clarity. Besides fixing these typos, consider improving the readability of the proofs by being a bit more explicit with "obvious" steps that may be currently omitted.
 - I think the set L of paired samples was not defined?
 - Line 59: "transformations identifies" --> "transformations identify"
 - Line 65: "samples available" --> "samples are available"
 - Lines 95-96: "the cross-modality samples share the same c are aligned" --> unclear meaning... maybe drop "share the same c"?
 - Line 117: "to met" --> "to be met"
 - Line 183: Sentence ends abruptly at: "where \Theta^(q)."

**Questions:**

1. What do you mean by "lift" the constraints in line 131?
2. Although theorem 1(a) does not "require" independence between c and p, isn't that implied/necessary? Otherwise, could you show that Dependence between p and c still yields identifiability of c? Specifically, say, if c1 and c2 are conditionally independent p(c1,c2,p) = p(c1|p)p(c2|p)p(p).
	2.a. Does this have to do with Line 149: Q^(q)A^(q) = [Θ^(q), 0] ?
3. Are there obvious limitations wrt differences in the sample size for x^(1) and x^(2)? Are there any provable biases if the data is highly unbalanced? How does unbalanced data affect the identifiability?
4. Is the methodology and identifiability theory limited to the two-dimensional case?

**Limitations:**

- A discussion of the asymptotic computational performance is amiss? Specifically, with respect to d_C and the total sample size.
- The paper does not appear to discuss the applicability of the theorems to the case of q > 2 (i.e., more than 2 modalities).
- Lacks a discussion of the stability of GAN training, especially with several additional loss augmentations.

---

> ### Author Rebuttal · Authors · 2024-08-07
>
> **[Code Clarity]** We will clean the code and change the variable names according to the notation used in the paper.  We found that Fig. 5 sometimes could not be replicated due to occasional failure of GAN convergence. We fixed the issue by increasing the regularization parameter $\beta$ from 0.001 to 0.005.
>
> To ensure reproducibility, we have fixed the random seed in the code. If the AC allows, we could share an anonymous link of the code for experiment in Fig. 5 (the rebuttal policy does not allow any link in the replies) through the AC. Please let us know.
>
>
> **[Results under Various Sample Sizes]** Following the reviewer’s suggestion, we have calculated the Amari distance [R1] between $\widehat{\bf{\Theta}}^{(1)}$ and $\widehat{ \bf{\Theta}}^{(2)}$ for different number of samples of unpaired data for CCA and SCA. The data for two views were generated, by sampling the shared component of dimension ($D=2$) from VonMises distribution and private components from Gamma and Laplace distributions.
>
> As the sample size decreases, the Amari distance increases for unaligned SCA because the distribution matching is difficult only using a few samples. CCA does not really work under this setting as it needs aligned cross-domain samples. However, our unaligned SCA does not need aligned samples.
>
> **Table 1**: Amari distance between $\widehat{\bf{\Theta}}^{(1)}$ and $\widehat{ \bf{\Theta}}^{(2)}$.
>
> ---
> N &nbsp;&nbsp;&nbsp;&nbsp;&nbsp;&nbsp;&nbsp;&nbsp;&nbsp;&nbsp;&nbsp; | SCA &nbsp;&nbsp;&nbsp;&nbsp;&nbsp;&nbsp;&nbsp;&nbsp;&nbsp;&nbsp;&nbsp; | CCA
>
> ---
>
> 100,000 | 6.5 × 10^-3  | 0.677
>
> 10,000 &nbsp; | 5.5 × 10^-3 | 0.533
>
> 1,000 &nbsp;&nbsp;&nbsp;&nbsp;| 8.4 × 10^-3 | 0.352
>
> 100 &nbsp;&nbsp;&nbsp;&nbsp;&nbsp;&nbsp;&nbsp;| 4.2 × 10^-3  | 0.364
>
> 50 &nbsp;&nbsp;&nbsp;&nbsp;&nbsp;&nbsp;&nbsp;&nbsp;&nbsp;| 0.071&nbsp;&nbsp;&nbsp;&nbsp;&nbsp;&nbsp;&nbsp;&nbsp;&nbsp;&nbsp;&nbsp;| 0.313
>
> 20 &nbsp;&nbsp;&nbsp;&nbsp;&nbsp;&nbsp;&nbsp;&nbsp;&nbsp;| 0.298 &nbsp;&nbsp;&nbsp;&nbsp;&nbsp;&nbsp;&nbsp;&nbsp;&nbsp;&nbsp;| 0.402
> ***
> **[Typos, Definitions]** Thank you for your careful reading. We will fix the grammar and typos, and include more detailed definitions.
>
> **[Questions]**
> **Q1.** We meant that our reformulation (7) uses a regularization term $R(\bf{Q}^{(q)})$ to promote the constraint in Problem (6c). This operation ``lifts'' up the constraint that was below the objective function into the new objective function. We will make this clearer.
>
> **Q2:** No, the independence between $\bf{c}$ and $\bf{p}$ is not necessary. In fact, Theorem 1 did not use the independence between $\bf{c}$ and $\bf{p}$. However, Theorem 1(a) did use marginal independence of the components of $\bf{c}$, i.e., for $\bf{c}$ $=[c_1, …, c_{d_C}]^T$, $p(c_1, …, c_{d_C}) = \prod_{i=1}^{d_C} p(c_i)$.
>
> Note that conditional independence does not imply marginal independence, i.e., $p(c_1, c_2, \bf{p})$ $= p(c_1 | p) p(c_2 | \bf{p}) p(\bf{p})$ does not imply $p(c_1, c_2) = p(c_1) p(c_2)$. Hence, assumption in Theorem 1(a) may not be satisfied, and identifiability cannot be guaranteed.  However, it is still possible for $\bf{c}$ to be dependent upon $\bf{p}$ and satisfy Theorem 1(a), e.g., $p(c_1, c_2, \bf{p} )$ $= p(c_1) p(c_2 |\bf{p}) $$p(\bf{p})$, then $p(c_1, c_2) = p(c_1) p(c_2)$.
>
> The statement in Line 149, $\bf{Q}^{(q)}\bf{A}^{(q)} = [\bf{\Theta}^{(q)}, 0]$, is the goal of Theorem 1. In our proof, this goal is achieved without the assumption that $\bf{c}$ and $\bf{p}$ are independent.
>
> **Q3.** Our conjecture to both of the first two questions is “yes”, but we have not had analytical underpinning yet. Empirically, unbalanced data is not friendly to such unaligned multi-domain learning problems. Our identifiability does not consider sample sizes of $\bf{x}^{(1)}$ and $\bf{x}^{(2)}$, or differences thereof. The analysis is carried out in the limit of infinite data, i.e., assuming an exact solution to the proposed optimization problem---which is already a challenging analysis problem. Therefore, data imbalance conditions are not covered by the current analysis. For latent component analysis problems, e.g., ICA and nonlinear ICA, the vast majority of the literature have to consider the population case to simplify analysis. Nonetheless, finite sample analysis does exist [R2], but such analysis itself is highly nontrivial and might deserve a standalone study.
>
> **Q4.** The methodology and identifiability theory are not limited to the two-dimensional case. Note that the latent dimensions $d_C$ and $d_P$ can be much larger positive integers.
>
> **[Limitations - Q>2 Case]**  The algorithm can be easily extended to Q>2 cases. Nonetheless, it does require additional work to understand what benefits can be brought upon in terms of identifiability by extra domains.
>
>
> **[Limitations - Discussion on GAN stability]** In our experience, GAN training is sensitive to hyperparameter setting, and can fail occasionally for different random initializations. However, adding regularization (e.g., homogenous mixing case, weakly supervised case, and classification loss in domain adaptation) does seem to improve the training stability. We will add this discussion.
>
> ## References
>
> [R1] Amari et al., 1995. A New Learning Algorithm for Blind Signal Separation.
>
> [R2] Lyu et al., 2022. Understanding Latent Correlation-Based Multiview Learning and Self-Supervision: An Identifiability Perspective.

---

> > ### Comment · Reviewer_DxMx · 2024-08-12
> >
> > Thank you for your clarifications. I have some follow up questions below:
> >
> > Sample size assessment: The Amari distances are quite remarkable (very low) considering the sample sizes. Could you clarify how you estimate \Theta? I recommend reporting this result (e.g., median +/- std Amari values) for every experiment (assuming you can estimate \Theta). CCA result does not need to be reported.
> >
> > Q2: Can you add the note about conditional independence in a footnote?
> >
> > Q3: Can you provide any empirical evidence about how unbalanced data affects identifiability in the different scenarios investigated here?
> >
> > Q4: So the current theory is limited to Q = 2 domains, but can be extended. Could you elaborate further on the point about "benefits" from extra domains? Do you anticipate diminished returns from including extra domains?
> >
> > Complexity analysis: Could you add a discussion about the computational complexity of the proposed model? Do the memory/computation requirements grow linearly/quadratically/other w.r.t. d_C?

---

> > > ### Author Response · Authors · 2024-08-13
> > >
> > > **[Amari Distance Computation]**
> > >
> > > In our evaluation, we used $\bf{\hat{\Theta}^{(q)}} = \bf{Q}^{(q)} \bf{A}^{(q)}$$(1:d\_C)$, where $\bf{A}^{(q)}$$(1:d\_C)$ represents the first $d\_C$ columns of $\bf{A}^{(q)}$. Note that $\bf{Q}^{(q)}$ is our estimated linear operator, and  $\bf{A}^{(q)}$ is the ground-truth mixing system that is available for synthetic data experiments (we only evaluated Amari distance for the synthetic data on our previous reply).
> > >
> > > Another note is that (thanks to the above discussion) we realized that general matrix distances (such as Euclidean distance) could be a better fit for our case than the Amari distance. To see, recall that we have content identifiability if and only if
> > > $  \bf{Q}^{(q)} \bf{A}^{(q)}  = [\bf{\Theta}, \bf{0}]$. Hence, we need $\hat{\bf{\Theta}}^{(1)} = \hat{\bf{\Theta}}^{(2)} = \bf{\Theta}$. However, Amari distance is insensitive (invariant) to permutation and scaling, i.e., $\hat{\bf{\Theta} }^{(1)} = \bf{P} \Lambda \hat{\Theta}^{(2)}$ incurs zero Amari distance, where $\bf{P}$ and $\Lambda$ are any permutation and scaling matrices respectively. Hence, we present the Euclidean distance instead of the Amari distance. Additionally, we also report the $\\| \bf{Q}^{(q)} \bf{A}^{(q)}$$(d\_C+1 : d\_C + d^{(q)}\_P ) \\|_F$ which needs to be close to $\bf{0}$ for identifiability.
> > >
> > > Table 1: Numerical evaluation of identifiability $\\| \widehat{\bf{\Theta} }^{(1)} (1: d\_C) - \widehat{ \bf{\Theta} }^{(2)}(1:d\_C) \\|_{F} .$
> > >
> > > |N            | SCA   | CCA |
> > > | :- | :- | :- |
> > > |100,000 | 0.009  | 1.368|
> > > |10,000   | 0.007  | 1.544|
> > > |1,000     | 0.003  | 2.206|
> > > |100        | 0.032  | 1.755|
> > > |50          | 0.133  | 1.667|
> > > |20          | 1.462  | 1.522|
> > >
> > > &nbsp;
> > >
> > > Table 2: &nbsp; $1/2  \sum\_{q=1}^2 \\| \widehat{\bf{\Theta}}^{(q)} ( d\_C+1 : d\_C+d^{(q)}\_P) \\|\_{F}$.
> > >
> > > |N           | SCA    | CCA |
> > > | :- | :- | :- |
> > > |100,000 | 0.021  | 0.284|
> > > |10,000   | 0.034  | 0.279|
> > > |1,000     | 0.002  | 0.329|
> > > |100        | 0.043  | 0.368|
> > > |50          | 0.131  | 4.092|
> > > |20          | 0.747  | 0.755|
> > >
> > >
> > > We will follow the reviewer’s suggestion and add the new experiment (with mean and standard deviation) for the synthetic data experiments in the revised version.
> > >
> > > **Q2.** Yes. We will add a footnote about conditional independence in the main paper.
> > >
> > > **Q3.** Thanks for the suggestion. We have run the following experiment with unbalanced data.
> > > For the following experiment, the data for two modalities were generated, by sampling the shared component of dimension($D=2$) from VonMises distribution and private components from Gamma and Laplace distributions.  The number of samples in the first modality is fixed to 100,000 and the second view ranges from 10,000 to 10 samples.
> > >
> > > Table 3:  Performance of SCA on imbalance data based on following two metrics,
> > >
> > > **metric1** = &nbsp; $\\| \widehat{\bf{\Theta}}^{(1)} (1: d\_C) - \widehat{ \bf{\Theta}}^{(2)}(1: d\_C) \\|\_{F} $,
> > >
> > >  **metric2** = &nbsp; $1/2  \sum\_{q=1}^2 \\| \widehat{\bf{\Theta}}^{(q)} ( d\_C+1: d\_C+d^{(q)}\_P) \\|\_{F}$.
> > >
> > > | \# samples in modality 2 | metric1 | metric2 |
> > > | :- | :- | :- |
> > > |10,000         | 0.008  | 0.025 |
> > > |1,000           | 0.025  | 0.015 |
> > > |100              | 0.091  | 0.087 |
> > > |10                | 1.375  | 0.209 |
> > >
> > > We will include the above result (with mean and standard deviation) in the revised version.
> > >
> > > **Q4. [Possible Benefit of $Q \geq 2$ domains]**
> > >
> > > One foreseeable benefit of having more than two domains is that Assumption 1, when modified for $Q \geq 2$ domains, could be more relaxed. This is because modality variability in general is satisfied if at least two of the total number of domains satisfy the current Assumption 1. Having more modalities can make the chance of Assumption 1 increased. On the other hand, enforcing the distribution matching constraint Eq. (6b) could be more challenging for more than two domains. We will add this discussion in the revised version.
> > >
> > > **[Complexity Analysis]**
> > >
> > > The short answer is that **both the memory and computational complexities of the proposed method scales linearly with** $d_C$. The per-iteration computational complexity is $O(B d\_C (d^{(1)} + d^{(2)}) )$, where $B$ is the mini-batch size. The per-iteration memory complexity is $O(B d\_C (d^{(1)} + d^{(2)}) )$ as well. The complexities are based on the fact that we use mini-batch based stochastic gradient-type optimizer. We will add a section in the appendix to detail the complexity calculation.

---

> > > > ### Comment · Reviewer_DxMx · 2024-08-13
> > > >
> > > > Thank you for your responses.
> > > > Conditioned on the inclusion of all adjustments, edits, results, and discussions the authors have committed to, I've updated my rating on Soundness from Good --> Excellent, largely due to the additional experiments and evidence provided.
> > > > Keeping my current score rating of 7 (Accept) as my overall impression of the work remains unchanged.

---

> > > > > ### Author Response · Authors · 2024-08-13
> > > > >
> > > > > We thank the reviewer for the constructive comments and discussion. We will make the promised changes as discussed.

---

### Author Rebuttal · Authors · 2024-08-07

### [ **Overall Response** ]

We sincerely thank all the reviewers for their effort in reviewing our manuscript.  Our responses are summarized as follows:

**Reviewer DxMx** suggested improving code clarity and observing the effect of sample size with a new evaluation metric. Following the comments, we ran an additional experiment and evaluated the Amari distance between $\widehat{\bf{\Theta}}^{(1)}$ and $\widehat{ \bf{\Theta}}^{(2)}$. We also improved our code readability and clarified the questions of the reviewer regarding the methodologies and identifiability.

**Reviewer 1Nhs’s** major comments were regarding the use of pre-processed data, theory-experiment consistency, significance of linear mixture models, and clarifications on our assumption and theorem. We provided clarifications. In particular, we pointed out that features obtained after common pre-processing tools (e.g., pre-trained models and word embeddings) are more likely to follow the linear mixture model by empirical evidence from the literature. Hence, such settings are arguably suitable for applying our methods.

**Reviewer tDhk** suggested discussing differences from existing content-style approaches. There are also comments regarding the practicality of the hyper-rectangle support assumption for the content and the use of the data processing inequality in our proof step. Following the suggestions, we explained use cases where the content could be seen as hyper-rectangle support. Further we clarified our proof step where we use data processing inequality and added the discussion regarding the connection with the previous related works.

**Reviewer YjhB** suggested using the same pre-processing for all methods in experiments. The reviewer also made a good point that using CLIP might not be appropriate as CLIP may have seen all the data under test. To address this concern, all methods now use ImageNet1k pretrained ResNet features instead of the CLIP features. The results are attached in the **enclosed PDF**. We have also added more recent baselines following reviewer suggestions.

&nbsp;

Due to space constraints for the rebuttal, we have the references used in the rebuttal are as follows:

&nbsp;

### **References**

[R1] Amari et al., 1995. A New Learning Algorithm for Blind Signal Separation.

[R2] Lyu et al., 2022. Understanding Latent Correlation-Based Multiview Learning and Self-Supervision: An Identifiability Perspective.

[R3] Common et al., 1994. Independent Component Analysis, A New Concept?

[R4] Lee et al., 1999. Learning the parts of objects by non-negative matrix factorization.

[R5] Erdogan et al., 2013. A Class of Bounded Component Analysis Algorithms for the Separation of Both Independent and Dependent Sources.

[R6] Huang et al., 2018. Multimodal Unsupervised Image-to-Image Translation.

[R7] Bhalla et al., 2024. Interpreting CLIP with Sparse Linear Concept Embeddings (SpLiCE).

[R8] Radford et al., 2021. Learning Transferable Visual Models From Natural Language Supervision.

[R9] Chen et al., 2020. A Simple Framework for Contrastive Learning of Visual Representations.

[R10] Cao et al., 2018. Joint profiling of chromatin accessibility and gene expression in thousands of single cells.

[R11] Joulin et al., 2016. FastText.zip: Compressing Text Classification Models.

[R12] Lample et al., 2018. Word Translation without using Parallel Data.

[R13] Yang et al., 2021. Multi-domain translation between single-cell imaging and sequencing data using autoencoders.

[R14] Kugelgen et al., 2021. Self-Supervised Learning with Data Augmentations Provably Isolates Content from Style.

[R15] Lyu et al., 2022. Understanding Latent Correlation-Based Multiview Learning And Self-Supervision: An Identifiability Perspective.

[R16] Gulrajani et al., 2022. Identifiability Conditions for Domain Adaptation.

[R17] Xie et al., 2023. Multi-Domain Image Generation And Translation With Identifiability Guarantees.

[R18] Sorensen et al., 2021. Generalized Canonical Correlation Analysis: A Subspace Intersection Approach.

[R19] Sturma et al., 2023. Unpaired Multi-Domain Causal Representation Learning.

[R20] Kong et al., 2022. Partial disentanglement for domain adaptation.

[R21] Choi et al., 2019. StarGAN v2: Diverse Image Synthesis for Multiple Domains.

[R22] Wu et al., 2018. Multimodal Generative Models for Scalable Weakly-Supervised Learning.

[R23] Rangwani et al., 2022. A Closer Look at Smoothness in Domain Adversarial Training.

[R24] Zhang et al., 2023. Free Lunch For Domain Adversarial Training: Environment Label Smoothing.

[R25] Goodfellow et al., 2014. Generative Adversarial Nets.

&nbsp;

&nbsp;

---

### Decision · Program_Chairs · 2024-09-25

**Decision:**

Accept (poster)

**Comment:**

The submission proposes a new approach for unaligned shared component analysis, where the multiple data modalities are drawn from a common Bayesian linear model with some latent shared components but are not aligned (or registered) to coincide sample by sample.

The contribution consists both in a novel algorithm minimizing a min-max distribution matching criterion somewhat inspired by GAN ideas and in sets of conditions under which such unaligned factor models can still be identifiable (up to fundamental permutation and scale indeterminacy). The significance of the problem that is addressed as well as the value and originality of both of these contributions have been recognized by the reviewers. The reviewers however made many relevant comments about weaknesses of the original submission that need to be taken into account by the authors. This is all the more important that several reviewers have raised their score following the discussions, taking into account the responses of the authors, including some additional experimental results.

Given the target selectivity set for the conference, the submission is close to the decision threshold but in light of priority given to the novelty of the content and of the clear path for improving the submission based on the exchanges with the reviewers, I am leaning toward acceptance at the conference.